# IndiMathBench: Autoformalizing Mathematical Reasoning Problems with a Human Touch

## Abstract

Reliable autoformalization remains an elusive goal even in the era of large language models (LLMs). Even the best LLMs struggle to translate natural language into formal constructs in languages like Lean. High-quality data has been a key bottleneck given the resource costs associated with manual curation and validation of these translations. On these lines, we introduce IndiMathBench, a human-verified benchmark designed to evaluate mathematical theorem proving, curated using an AI-powered human-assisted pipeline for formalizing natural language problems in Lean. IndiMathBench is composed of 312 formal Lean4 theorems paired with their corresponding informal problem statements, sourced from Indian Mathematics Olympiads. Our pipeline uses category-based retrieval and a self-debug loop with feedback from a symbolic checker to generate candidate formalizations. Multiple such formalizations are generated using an ensemble of LLMs. These formulas are presented with a summary to the human through an interactive dashboard. The dashboard enables efficient validation and repair by the human. We analyze the performance of several state-of-the-art models on IndiMathBench, which will facilitate further research on automated theorem proving.

## 1 Introduction

The formalization of mathematical knowledge has long been a central goal in computer science and mathematics, promising to enable mechanized proof verification, automated theorem proving, and systematic knowledge organization. Beyond theoretical interest, formalized mathematics offers practical benefits including error detection in published proofs, construction of reliable mathematical software, and development of intelligent tutoring systems that can provide rigorous mathematical guidance. Recent breakthroughs in Large Language Models (LLMs) have renewed interest in autoformalization (Bansal & Szegedy, 2020; Agrawal et al., 2022a; Gadgil et al., 2022; Wu et al., 2022a), the automatic translation of informal mathematical statements into formal logical representations, as a pathway toward more capable mathematical reasoning systems.

Despite significant progress in model capabilities, the evaluation of autoformalization remains fragmented and limited in scope. Existing benchmarks suffer from several critical limitations that hinder our understanding of true model performance and progress in the field. First, many benchmarks focus on popular competition problems from sources like the International Mathematical Olympiad (IMO) or Putnam Competition, which increasingly suffer from data contamination as they become integrated into large-scale model training datasets (Jiang et al., 2024). This contamination makes it difficult to distinguish between genuine mathematical reasoning and memorization of training examples. Second, creating high-quality benchmarks requires substantial manual effort from experts in both mathematics and formal verification systems, involving careful annotation and validation of formalizations (Yu et al., 2025). Third, current evaluation frameworks often employ simplistic binary metrics (correct/incorrect) and fail to provide fine-grained analysis of different types of formalization errors, making it challenging for researchers to identify specific areas for model improvement.

We address these limitations by introducing IndiMathBench , a benchmark for mathematical autoformalization built from Indian Mathematical Olympiad problems. Our benchmark contains 312 carefully curated problems spanning diverse mathematical domains, geometry, algebra, number theory, and combinatorics, each paired with human-verified Lean 4 formalizations. Wxe conducted

systematic human verification of all formalizations, ensuring high-quality ground truth for reliable evaluation. A sample benchmark is shown in Figure 1.

Our key contributions are:

- A novel formal theorem proving benchmark set in Lean4, created with help of LLMs and verified by humans.
- A Visual Studio Extension to improve human-AI collaboration for Lean annotations.
- A framework for formalization that uses category-based retrieval and self-debug loop with a Lean validator, and use of an ensemble of LLMs.
- Analysis on different frontier foundational models on their Autoformalization Capabilities.

## 2 RELATED WORK

**Formal Theorem Proving Benchmarks** The evaluation of automated theorem proving systems has relied heavily on formal benchmarks, yet these resources exhibit significant limitations in scale and diversity. Early benchmarks like TPTP (Sutcliffe, 2017) and Mizar Mathematical Library (Grabowski et al., 2010) established foundational evaluation frameworks, while MiniF2F's 488 competition problems became a standard for modern formal mathematics evaluation (Zheng et al., 2021). Subsequent efforts expanded coverage: Putnam-Bench added 640 undergraduate problems (Tsoukalas et al., 2024b), ProofNet introduced 371 undergraduate-level theorems (Azerbayev et al., 2023a), and MATH dataset provided 12,500 competition problems (Hendrycks et al., 2021), though primarily in natural language.

Let n be a natural number. Prove that:

$$\left\lfloor \frac{n}{1} \right\rfloor + \left\lfloor \frac{n}{2} \right\rfloor + \left\lfloor \frac{n}{3} \right\rfloor + \cdots + \left\lfloor \frac{n}{n} \right\rfloor + \left\lfloor \sqrt{n} \right\rfloor$$

is even.

```
theorem inmo_2014_2 (n : ℕ) :
  Even
    ((Finset.sum (Finset.range n)
      fun i => ⌊(n : ℝ) /
        ((i + 1) : ℝ)⌋)
    + ⌊Real.sqrt n⌋) := by
  sorry
```

Figure 1: A formalization of INMO 2014 problem 4 in Lean 4

Cross-system portability has improved with MiniF2F extensions to Metamath, Isabelle, and HOL Light, while specialized benchmarks like CoqGym (Yang & Deng, 2019) and LeanStep (Han et al., 2022) focus on proof step prediction. However, these benchmarks predominantly reflect Western mathematical competitions (AMC, AIME, IMO, Putnam) and undergraduate curricula, underrepresenting diverse mathematical traditions and advanced research areas (Wu et al., 2022b; Polu & Sutskever, 2020). Recent efforts like FrontierMath push toward research-level problems where leading models achieve less than 2% success rates (Collaboration, 2024), while others explore domain-specific evaluation in areas like algebraic topology (Avigad et al., 2022) and category theory, highlighting the substantial gap between current capabilities and human mathematical reasoning across diverse mathematical domains.

**Autoformalization with Large Language Models** The translation of informal mathematics to formal specifications has seen rapid progress through large language models. Early work by (Wu et al., 2022b) demonstrated 25.3% success on competition problems using few-shot prompting, despite formal mathematics comprising only 0.18% of pretraining data. (Agrawal et al., 2022b) achieved 75% accuracy on undergraduate theorem statements, while (Jiang et al., 2023b) introduced the Draft, Sketch, and Prove methodology for mapping informal proofs to formal sketches. Recent systems have shown substantial improvements: DeepSeek-Prover achieves 46.3% accuracy on Lean 4 MINIF2F (AI, 2024), while other approaches have explored fine-tuning strategies (Azerbayev et al., 2023b), reinforcement learning from proof assistant feedback (Polu & Sutskever, 2020), and neural-symbolic integration (Han et al., 2022). Despite these advances, fully automated approaches continue to struggle with semantic consistency, complex mathematical reasoning, and the domain gap between natural language and formal specifications (Zheng et al., 2022; Welleck et al., 2021; Kirtania & Iyer, 2025). The field has increasingly moved toward hybrid human-AI methodologies that combine automated translation capabilities with human expertise in both competition mathematics and theorem proving languages (Cohen et al., 2023; Bansal et al., 2019). Current bottlenecks remain the requirement for semantic and syntactic correctness, the time-intensive nature of expert

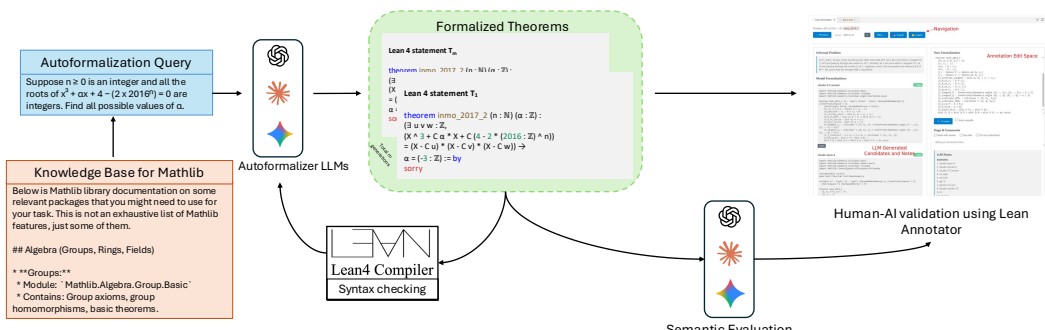

Figure 2: Overview of approach for creating IndicMath dataset: Human is assisted by a multiple LLM annotators. Each LLM generation is conditioned on the natural language and documentation, and goes through a validation check by Lean. Errors are provided as feedback in subsequent iterations. The final generations from all LLMs are summarized by an LLM in a dashboard to help optimize annotation efficiency.

annotation, and the scarcity of large-scale parallel datasets linking informal mathematical statements to their formal counterparts.

**Human-AI Collaboration in Formalization**

The formalization of research-level mathematics increasingly relies on human-AI collaboration. Tao's real-time formalization of the Polynomial Freiman-Ruzsa conjecture demonstrated that cutting-edge results can be formalized through collaborative efforts, though with a 25x reduction in working speed (Tao, 2023). The LeanDojo platform provides 122,517 theorems with fine-grained annotations for training neural provers (Yang et al., 2023). Process-supervised approaches use formal system feedback to improve translation quality while reducing annotation requirements. These hybrid methodologies consistently outperform pure automation by combining human expertise for conceptual insights with AI capabilities for pattern recognition and verification.

**Evaluation Methodologies**    Evaluation of autoformalization has evolved from syntactic metrics like BLEU scores to semantic assessment approaches. The Generalized Tree Edit Distance (GTED) computes structural similarity through operator trees (Liu et al., 2025a), while Bidirectional Extended Definitional Equivalence (BEq) provides neural-symbolic checking with 90.5% accuracy on expert data (Liu et al., 2025b). FormalAlign combines certainty and similarity scores through contrastive learning (Lu et al., 2024). However, scalable evaluation remains challenging, with manual expert verification being costly but necessary for establishing reliable ground truth. The correction effort scale (0-4) provides standardized quality assessment based on human repair effort (Jiang et al., 2023a).

## 3    INDIMATHBENCH

IndiMathBench is a human-validated evaluation benchmark set consisting of 312 formalized problem statements from Indian Olympiad problems. This benchmark set is prepared using a hybrid human-AI pipeline designed for maximally reducing the human effort required for formalizing benchmarks. Each benchmark consists of a description (in English) of a math problem, a Lean4 (de Moura et al., 2015) theorem corresponding to that problem, and any numerical solution where applicable. The English descriptions are sourced from the Regional Mathematical Olympiad (RMO) and Indian National Mathematical Olympiad (INMO) examinations in India.

| Category | Count |
|---|---|
| Geometry | 98 |
| Algebra | 92 |
| Set Theory & Combinatorics | 45 |
| Number Theory | 77 |

Table 1: Problem counts by topic domain in IndiMathBench. The distribution is representative of a typical RMO or INMO paper.

The RMO and INMO are used to select India's most promising high-school student. Students who pass the RMO qualify to take the INMO, which is a significantly more difficult national-level test. The Human-AI collaborative process is described more in Section 4.

**Diversity and Breadth.** Compared to existing benchmarks like MINIF2F (Zheng et al., 2021), which include a wide selection of high-school and early undergraduate mathematics problems (primarily from AMC, AIME, and IMO), the IndiMathBench focuses on problems from the Indian Mathematical Olympiad system. These problems are drawn from national and regional competitions in India, and are restricted to the high-school curriculum, covering algebra, number theory, geometry, and combinatorics. Unlike MINIF2F, which incorporates problems involving topics such as inequalities, calculus, or matrix algebra, INMO/RMO problems exclude calculus and typically avoid higher-level abstractions.

Despite this narrower domain scope, INMO problems in particular demonstrate high internal diversity and depth. Many problems involve multi-step reasoning, uncommon constructions, and non-standard techniques. For example, geometric problems often require diagrammatic insight combined with multiple auxiliary constructions, while number-theoretic questions tend to involve clever use of parity, bounding, or invariant arguments.

**Problem Domains.** IndiMathBench problems are traditionally sourced from a fixed set of topics: algebra, euclidean geometry, elementary number theory, and combinatorics. Calculus, set theory, and linear algebra are not part of the official syllabus. This restriction makes the benchmark more uniform in scope, but allows for deeper exploration of problem-solving within each domain. For example, geometry problems frequently involve classical triangle centers (e.g., orthocenter, centroid) or cyclic quadrilaterals, which require nontrivial formalization in Lean.

**Formalization Effort.** The formalization was done over the course of a month by two annotators with moderate level of expertise in using lean for formal proof writing. AI was used extensively throughout the process and we discuss that in depth in 4. Some problems (36%) in our set include having to solve for a value rather than proving a condition. We take a similar approach as MINIF2F for this case by re-framing the question "solve Question Q for Solution X" as a "prove Question Q iff Solution X".

**Comparison with Existing Benchmarks.** While MINIF2F provides more variety in domain coverage, especially by including undergraduate topics, the IndiMathBench emphasizes conceptual depth in classical problem areas. This makes it a useful complement to existing formal benchmarks, particularly for studying formalization strategies for diagrammatic reasoning, geometric constructions, and informal-to-formal translation in constrained domains.

## 4 TECHNIQUE : AUTO-FORMALIZATION APPROACH

We describe our scalable approach for leveraging general purpose LLMs to maximize human annotation efficiency in the formalization of mathematical problems. This is the approach we used for generating IndiMathBench. Our approach is general and has some salient features and reusable components. The approach is designed to maximize annotation efficiency for human experts. The key features of our approach are (a) category-based retrieval, (b) feedback loop using symbolic validation, and (c) multi-model generation and consolidation. We start with a set $\mathcal{P}$ of math problems in English. For each problem $p \in \mathcal{P}$, the final result is a custom dashboard for the human annotator that contains multiple formalizations from various models, their validation status, and a textual summary. Figure 3 depicts a screenshot of the dashboard.

### 4.1 AUTOMATED FORMALIZATION GENERATION

In our initial evaluations of LLMs for autoformalization, the main deficiency observed was the poor quality of formulas written in custom formal languages. The difficulty is characterized by a tendency to hallucinate content, such as non-existent imports, and to confuse or mix up syntax from various other theorem proving languages or lean 3. We observe giving access to snippets of library code and documentation as feedback helps the LLM generate well-formed formulas. This is especially useful in geometry theorems where the mathlib library does not natively support a lot of operations typical of competition-level geometry problems. It is this fundamental inability to adhere consistently to a custom formal syntax that directly motivates the need for a more structured process that incorporates documentation access and feedback to help the LLM generate well-formed, syntactically correct formulas.

---

**Algorithm 1** LLM-Based Autoformalization of Mathematical Problems

---

```
 1: procedure FORMALIZE(p, Model)                1: procedure PREPROCESS(P, Categories)
 2:              ▷ p: problem description in NL    2:                  ▷ P is a set of problems
 3:     ctxt ← p.Cat.Ctxt                          3:     for all p ∈ P do
 4:     f ← MODEL(p, ctxt)                          4:         p.Cat ← Label(p, Categories)
 5:     for i = 1 to 6 do                           5:     end for
 6:         errors ← VALIDATEINLEAN(f)             6:     for all cat ∈ Categories do
 7:         if errors = ∅ then                      7:         cat.Samples ← Sample(P, cat)
 8:             break                               8:         cat.Ctxt ← Retriever(cat.Samples)
 9:         end if                                  9:     end for
10:         feedback ← PARSEERRORS(errors)        10: end procedure
11:         f ← MODEL(p, ctxt, f, feedback)       11: procedure POSTDISPLAY(p)
12:     end for                                    12:     F ← {}
13:     return f                                    13:     for all model ∈ ModelList do
14: end procedure                                  14:         F.Add(FORMALIZE(p, model))
15: procedure MAIN(P, Categories)                 15:     end for
16:     PREPROCESS(P, Categories)                  16:     summary ← MODEL(F, p)
17:     POSTDISPLAY(p) for p ∈ P                   17:     return F, summary
18: end procedure                                  18: end procedure
```

---

The pseudocode for our general approach to go from a problem to its dashboard for human oversight is shown in Algorithm 1. The approach can be broken down into three key steps:

(1) *Category-Based Retrieval*: In the preprocessing step, we retrieve context to be used during formalization of each problem. There is one static context retrieved for each *category* of problems.

(2) *Iterative Refinement with Error Feedback*: In the second step, the LLM generates a formula conditioned on the given informal problem statement and the context, and then iteratively refines the generated formula based on feedback from errors detected by the formal tool (Lean4 validator).

(3) *Multi-Model Ensemble and Comparative Analysis*: In the third step, the final generations from multiple LLMs are collected and presented to the user in a dashboard along with validation results and a summary (generated by LLM). The user generates the final formula using this information.

**Step 1 Preprocessing: Category-based Retrieval.** General-purpose LLMs often struggle with getting the right imports, notation and type conversions while writing in a low-resource language like Lean. Recognizing that mathematical formalization requires deep knowledge of existing libraries and conventions, we augment our prompts with an automatically curated Mathlib documentation. The preprocessing step is mentioned in Procedure `PreProcess` in Algorithm 1. We first use an LLM (with a final human supervision) to label all the problems in the problem set $\mathcal{P}$ with a label from the set `Categories` in Table 1. We then randomly sample 25% of the problems from each category. We then use an LLM agent – built using claude-sonnet-4 – that has bash access to files within the mathlib library repository. The agent is tasked to explore the vast library and extract the definitions and formulas that may be most relevant to the task of formalizing the problems given to it. For each category in `Categories`, we invoke this agent and give it the sampled problems for that category. The agent returns the *static context* to be used for that category. This static context provides essential domain knowledge needed for formalizing problems from that category without hallucinations and syntactic errors. The prompt for the retrieval agent can be found in Appendix A.3.1.

**Step 2 Main Loop: Iterative Refinement with Error Feedback.** Figure 2 provides a high-level overview of our main loop for formalizing problems. Specifically, as shown in Procedure `Formalize` in Algorithm 1, the system operates as follows: (1) Generation: First the LLM generates a formalization $f$ for the informal problem statement $p$ conditioned on the static context extracted for the category of problem $p$. See the prompt in Appendix **??**. There is a slight nuance here for "solve"-type problems, where we also include the solution and prompt the LLM to write formula that verifies the solution. (2) Validation: Next the generated theorem is validated using the Lean 4 compiler, `ValidateInLean`, which produces actionable error messages if the theorem fails compile check (using an external `ParseErrors` function). (3) Refinement: The LLM is prompted to fix its errors given the error messages. This refinement process is repeated for a maximum of 6 iterations. Using this iterative procedure helps us achieve a 95.3% coverage; i.e., at least one model generation compiles successfully for 95.3% of the problems.

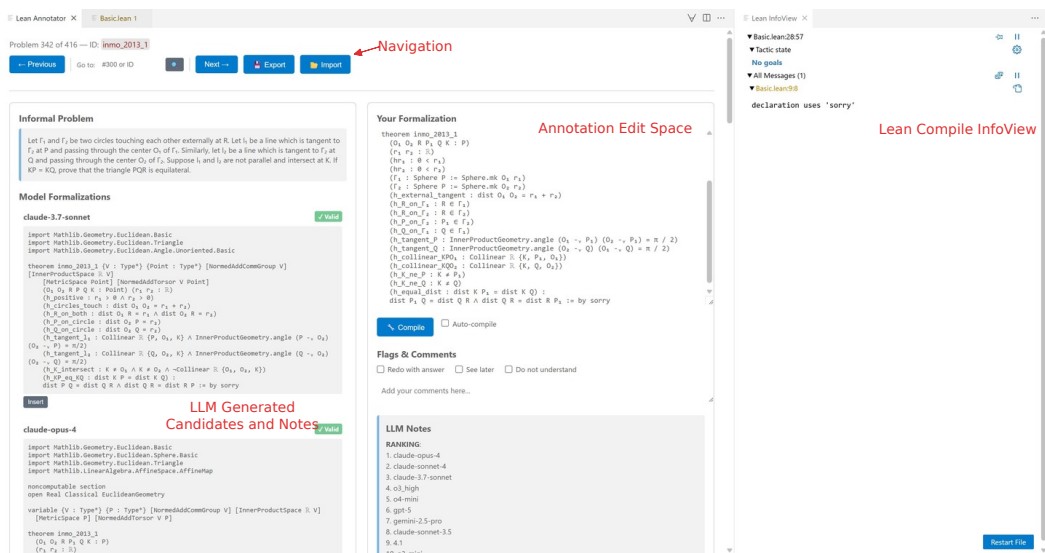

Figure 3: The Annotation Dashboard for the human expert to analyze various ranked options, modify and validate the final entry.

**Step 3: Multi-Model Ensemble and Comparative Analysis.** As described in Procedure `PrepDashboard` in Algorithm 1, each problem $p$ is processed by 12 state-of-the-art language models simultaneously, including GPT variants, Claude models, and their reasoning family of models (see figure performance). This ensemble approach serves multiple purposes: it provides redundancy against model-specific failures, enables comparative analysis of different architectural approaches to mathematical reasoning, and increases the likelihood that at least one model produces a correct formalization for each problem. The aggregated generations across the different models are then summarized by an LLM (GPT-5). The summary contains rankings for different models based on the problem statement, what each generation got wrong and others got right, and any correctness and completeness issues with the formalizations. We choose different models as an exercise to later evaluate their autoformalization capabilities, but the process can also be effective for multiple generation with the same model, analogous to self-consistency.

### 4.2 HUMAN-AI COLLABORATIVE ANNOTATION DASHBOARD

Figure 3 illustrates the comprehensive dashboard employed for the efficient human expert review of the final generations from our computational pipeline. This interface integrates all model outputs, corresponding validation results, and automated quality assessments and summaries over the group of formalizations as a whole. The dashboard also incorporates features that the VS Code Lean extension supports, with an option to compile as needed directly. A key feature is the inclusion of AI-generated annotations, computed via the `summary` variable in Procedure `PrepDashboard` – providing reviewers with comparative analyses of different formalization approaches, identification of common patterns or errors, missed out conditions, and preliminary quality assessments based on compilation success and semantic correctness.

These synthesized insights reduce the cognitive load on human annotators, allowing them to maintain the critical human oversight necessary for mathematical accuracy while concentrating their efforts on genuinely ambiguous cases rather than routine validation. Furthermore, the dashboard presents results from different large language models (LLMs), enabling human annotators to consolidate and refine outputs—for instance, by integrating a missing condition from one generated formalization into another. This capability provides human experts with a high-quality initial formulation that can be quickly refined into the final, correct formula, thereby enabling the scalable creation of high-quality formal mathematics benchmarks with minimal human effort.

The dashboard, prioritizes display of verified results—those that successfully compile and pass basic correctness checks—while still providing access to failed attempts and their error traces for com-

prehensive analysis. This design supports both rapid annotation workflows for "easy" cases and detailed investigation for "hard" cases of mathematical formalization.

We release this dashboard as a VS Code extension to the community, and it can be generally used for informal to formal, and formal to formal tasks in other languages as well. We hope this extension can serve the wider community create datasets faster.

### 4.3 ANNOTATOR EFFICIENCY STUDY

We conducted a small controlled study to evaluate the impact of LLM-generated candidates and group comparison summaries on informal to formal annotation efficiency for our case. The study compares three workflow variants:

1. **Full System**: provides candidate generations, automated summary notes, and a human-in-the-loop dashboard. This is our proposed system.
2. **Masked Candidates**: all candidate generations are displayed, but model identities are hidden and no summary notes are provided, simulating how much help multiple formalizations can give even without having an LLM group critique parts over a manual workflow.
3. **Manual Formalization**: the annotator formalizes from scratch without any pre feeded LLM assistance.

The three sets are drawn from three different consecutive years, with each set comprising 12 RMO problems. This design ensures consistent question styles within each set while providing coverage across problems with similar difficulty. A single annotator formalized three sets across each workflow. The annotator is allowed full internet and AI Assistant access throughout every study. We record the total time spent and the number of problems successfully formalized for comparison.

Figure 4 compares the time spent for over annotations across the three workflow variants. The manual formalization took an average of 14 minutes per problem. This is in line with PutnamBench's Tsoukalas et al. (2024a) average of 25 minutes and MINIF2F's Zheng et al. (2021) 10 minutes per problem. Compared to that, using only multiple masked LLM generated formalizations took 9 minutes per problem, a 60% speed up over manual. The annotator attributes this to mainly not having to write the problem structures themselves, and having to just verify and replace parts where one generation got some part wrong and some part right. The annotator also spent 32 minutes on a single particularly difficult problem which none of the generations had a close formalization for, and the annotator spent much of the time navigating the mathlib

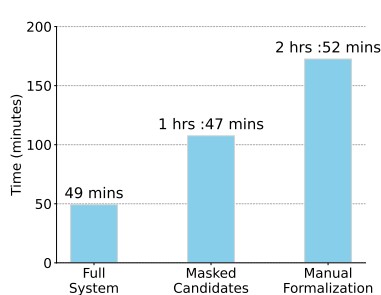

Figure 4: Annotation time for 12 RMO problems under the three workflows

library. With our full system, the average time spent formalizing was 4 minutes per problem. This is about 3.5x faster (251% speed up) than manually doing it and 2.2x faster (118% speed up) over the one without summary. The annotator notes that most times if there was something wrong with the best ranked generation in the summary, the summary would point it out and they would find the same part well-written in another generation. An example showcasing this speed up is noted in 8.

## 5 EXPERIMENTAL RESULTS

### 5.1 AUTOFORMALIZATION EVALUATION

**Setup.** As described in 4.1, we generate candidate formalizations, given a natural language problem statement, across 12 different general-purpose frontier LLMs. These include Claude Sonnet 4, Claude Opus 4, o3 (high), GPT-4.1, GPT-5, and Gemini 2.5 Pro, among others. Here, we aim to measure how semantically close the generated formalizations are to the human annotation.

**Evaluation Metrics** Evaluating autoformalization quality presents unique challenges due to the rigorous logical nature of formal mathematical statements, where seemingly minor syntactic varia-

tions can have significant semantic implications. To provide a comprehensive assessment, we employ two complementary evaluation metrics that have demonstrated high inter-annotator agreement with human evaluations.

**Bidirectional Equivalence (BEq)** (Liu et al., 2025b) evaluates logical equivalence by attempting to prove each theorem using the other as a premise. Specifically, given two Lean 4 theorems in sorry-format, `theorem_A` and `theorem_B`, BEq employs a diverse set of tactics both heuristically-driven and LLM-guided—to establish proofs in both directions. The formalization is deemed correct only if both directional proofs succeed, ensuring true logical equivalence rather than mere syntactic similarity.

**Generalized Tree Edit Distance (GTED)** (Liu et al., 2025a) provides a syntactic similarity measure by constructing operator tree representations of Lean theorems and computing the minimum cost of tree transformations required to convert one representation to another. GTED produces normalized scores between 0 and 1, where higher values indicate greater structural similarity.

These metrics are fundamentally contrasting in their approach: BEq captures semantic equivalence through the Lean proof engine's logical reasoning capabilities, while GTED quantifies syntactic structural similarity through tree transformations. This complementary evaluation framework provides both logical validation and structural analysis, offering a complete assessment of autoformalization quality. For our evaluation, we use Claude Sonnet 4 (temperature 0.7) with pass@1 for BEq

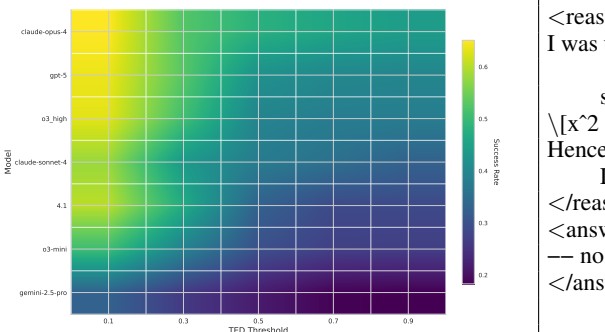

<reasoning>
I was unable to discover (within the time available) a
    satisfactory infinite−family of integer triples
    satisfying
$$[x^2 + y^2 + z^2 = (x − y)(y − z)(z − x).]$$
Hence I cannot at this moment supply a complete
    Lean proof of the required proposition.
</reasoning>
<answer>
−− no code
</answer>

Figure 5: Heatmap for GTED based on thresholds. Better models see a lighter hue.

Figure 6: o3 refuses to write a code after a few minutes of thinking on medium reasoning.

calculations, deviating from the original implementation's use of InternLM (Ying et al., 2024). All other implementation details follow the original specifications.

**Results.** Table 2 depicts the models evaluated against the BEq and GTED metrics. The model Claude Opus 4 does best across metrics. Figure 7 depicts the overlap on the benchmarks that Claude Opus 4, Gemini 2.5 Pro and GPT-5 solve (based on BEq metric). This shows that the models have certain complementary abilities and justifies our ensemble approach. In 160 of the problems, at least one generation passed the BEq check, i.e. for 51.2% of our dataset an LLM had formalized the problem correctly. Another notable detail is that among the three models, Claude Opus 4, Gemini 2.5 Pro and GPT-5, with a cumulative BEq passing for 108 problems, only 12 were from Geometry. This highlights the LLM's difficulty with using Mathlib's lacking support for Olympiad style geometry.

## 5.2 INDIMATHBENCH EVALUATION

We first compare our `Formalize` procedure in Algorithm 1 with a zero-shot baseline generation. In **zero-shot** baseline generation, the LLM is prompted to convert natural language mathematical problems into Lean 4 formalizations, but without giving it access to any additional context and without iterative feedback. We do this comparison for multiple state-of-the-art pre-trained language models. The results of the comparison between `Formalize` and the zero-shot baseline are shown in Table 3. Note that the table counts the number of problems that were "successfully" formalized by the two approaches. Here, we declare "success" if the generated formula passes validation by Lean. We note here that this is only a partial notion of correctness of a formula that is necessary,

| Model | ZS | KB+FB |
|---|---|---|
| GPT-4.1 | 58 | 132 |
| Claude Opus 4 | 17 | 310 |
| Claude Sonnet 4 | 15 | 272 |
| o3 (high) | 92 | 263 |
| GPT-5 | 127 | 297 |
| Gemini 2.5 Pro | 16 | 184 |

Table 3: The number of Lean-validated formulas generated by the different models in zero-shot (ZS) setting and in the setting with documentation (KB) and feedback (FB) loops. Across all models, we get a total of 416 Lean-validated benchmarks (out of which 312 are further manually processed.)

| Model | Success Rate |
|---|---|
| GPT-4.1 | 0/312 |
| Gemini 2.5 Pro | 0/312 |
| Claude Sonnet 4 | 1/312 |
| GPT-5 | 1/312 |
| o3(medium) | 1/312 |

Table 4: Evaluation results of various frontier models on IndiMathBench. Success Rates here refer to Lean Verifiable proofs submitted by the models.

but not sufficient, for full correctness. The numbers in Table 3 clearly show that error feedback and documentation retrieval contribute to success *across all models*. Notably Claude models both start off at pretty low compile success at one turn, but it quickly refines itself from the knowledge available from the Lean environment as well as the KB available at disposal.

| Model | BEq (312) | GTED Mean (%) | #GTED >0.9 (312) | Compiled Successfully (312) |
|---|---|---|---|---|
| Claude-opus-4 | **67** | **0.512** | **138** | **243** |
| Claude-sonnet-4 | 54 | 0.419 | 103 | 215 |
| Gemini-2.5-pro | 44 | 0.236 | 58 | 151 |
| GPT-5 | 36 | 0.475 | 124 | 235 |
| O3_high | 30 | 0.457 | 119 | 205 |
| 4.1 | 33 | 0.392 | 90 | 120 |

Table 2: Comparing models across BEq, GTED mean, # of samples with a GTED score>0.9, and compilation validity counts.

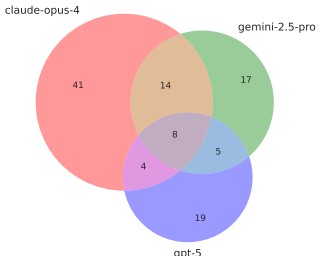

Figure 7: Venn diagram for the BEq passing problems for 3 models selected from different families.

### 5.3 AUTOMATED THEOREM PROVING EVALUATION

**Setup.** We measure the difficulty IndiMathBench poses to frontier general purpose LLMs for formal theorem proving, by testing them for completing the theorem statement with a valid proof.

**Results.** We evaluate each LLM on a single turn, pass@1 metric. We carry out these evaluations across the best frontier models available to us, Claude Sonnet 4, GPT-5, GPT-4.1, o3 (medium), and Gemini 2.5 Pro, at default parameters. Claude Sonnet 4, GPT-5, and o3 (medium) each resolve a single problem across the 312 strong IndiMathBench . This singular problem in all three was inmo_2015_5, where the solution simply involved using the pitot_theorem from Mathlib.

## 6 CONCLUSION

We introduce a new Lean 4 autoformalization benchmark (IndiMathBench) containing human verified Lean 4 formalizations of Olympiad level problems. Our comparative analysis shows that current frontier Large Language Models (LLMs) struggle with formal mathematics, passing only a single problem, which highlights the dataset's complexity. We also present a framework to ease manual annotation by utilizing documentation, compiler feedback, and aggregating multi-LLM generations. This approach is vital for low-resource formal languages and new codebases like LeanGeo (Song et al., 2025). The dataset and a VS Code dashboard extension are provided as an open resource to advance the field of neural theorem proving.

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

## A    EXAMPLE BENCHMARKS FROM INDIMATHBENCH

Consider the following problem statement:

> In a triangle ABC, let D be a point on the segment BC such that AB + BD = AC
> + CD. Suppose that the points B, C and the centroids of triangles ABD and ACD
> lie on a circle. Prove that AB = AC.

Its formalization in the benchmark set as follows:

```
variable {V : Type*} {P : Type*} [NormedAddCommGroup V]
  [InnerProductSpace ℝ V] [MetricSpace P] [NormedAddTorsor V P]

theorem inmo_2014_1 (A B C D : P)
  (hD : ∃ t : ℝ, 0 ≤ t ∧ t ≤ 1 ∧ D = AffineMap.lineMap B C t)
  (h_sum : dist A B + dist B D = dist A C + dist C D)
  (h_concyclic : Concyclic {B, C,
    centroid ℝ {A, B, D} id,
    centroid ℝ {A, C, D} id}) :
  dist A B = dist A C := by sorry
```

Consider the following problem description.

> Suppose $n \geq 0$ is an integer and all the roots of $x^3 + \alpha x + 4 - (2 \cdot 2016^n) = 0$
> are integers. Find all possible values of $\alpha$.

This is a problem that requires a solution. We assume a solution is given. For such cases, the formula
states that the given solution is actually correct.

```
import Mathlib.Data.Int.Basic
import Mathlib.Algebra.Polynomial.Basic

open Polynomial

theorem inmo_2017_2 (n : ℕ ) ( α : ℤ ) :
    ( ∃ u v w : ℤ ,
      (X ^ 3 + C α * X + C (4 - 2 * (2016 ^ n))
        = (X - C u) * (X - C v) * (X - C w)) ⇒
    α = (-3 : ℤ ) := by
  sorry
```

### A.1    EXPERIMENTAL SETUP

All formalizations and experiments were conducted using the Lean theo-
rem prover, version 4.22.   We relied on the `mathlib` library at commit
`f858fcc3b49c546705ba7d79c58217e85aaa5f0e` to ensure reproducibility and con-
sistency across our proofs and auxiliary results.

Our computational environment consisted of:

- **Hardware:** 8-core CPU, 32 GB RAM
- **OS:** Windows 11 Pro (64-bit)
- **Lean Toolchain:** Installed via `elan`, with `lake` for project management

All proofs were compiled and verified using Lean's native `lake build` system without additional
modifications to `mathlib`. The experiment scripts, proof files, and configuration details are pro-
vided in the supplementary material to facilitate full reproducibility.

### A.2    IMPACT OF LLM NOTES

In this section we discuss the impact of LLM generated notes to reduce effort of annotation. While
our primary focus is on improving proof generation through activation steering, we observe an ad-

```
import Mathlib

noncomputable section
open Classical Real EuclideanGeometry InnerProductGeometry

variable {V : Type*} {P' : Type*}
  [NormedAddCommGroup V] [InnerProductSpace ℝ V]
  [MetricSpace P'] [NormedAddTorsor V P']

theorem rmo_2013_4_1
    (Γ Λ : Sphere P') (O A B C D P : P')
    (hcenter : Γ.center = O)
    (hAΓ : A ∈ Γ) (hBΓ : B ∈ Γ)
    (hAΛ : A ∈ Λ) (hBΛ : B ∈ Λ) (hOΛ : O ∈ Λ)
    (hCΓ : C ∈ Γ) (hDΓ : D ∈ Γ) (hdiam : midpoint ℝ C D = O)
    (hPΛ : P ∈ Λ) (hPline : Collinear ℝ C D P) (hPO : P ≠ O)
    : angle (A -ᵥ P) (C -ᵥ P) = angle (B -ᵥ P) (D -ᵥ P) := by
  sorry
```

BEST FORMALIZATION:
gpt-5 provides the cleanest and most mathematically accurate formalization. It correctly models both circles as `Sphere P'` objects, properly handles the diameter condition with `midpoint ℝ C D = O`, uses appropriate collinearity constraints, and has clear, well-structured hypotheses without unnecessary complexity.

```
import Mathlib

noncomputable section
open Real EuclideanGeometry InnerProductGeometry

variable {V : Type*} {P' : Type*}
  [NormedAddCommGroup V] [InnerProductSpace ℝ V]
  [MetricSpace P'] [NormedAddTorsor V P']

theorem rmo_2013_4_1
    (Γ Λ : Sphere P') (O A B C D P : P')
    (hcenter : Γ.center = O)
    (hAΓ : A ∈ Γ) (hBΓ : B ∈ Γ)
    (hAΛ : A ∈ Λ) (hBΛ : B ∈ Λ) (hOΛ : O ∈ Λ)
    (hCΓ : C ∈ Γ) (hDΓ : D ∈ Γ) (hdiam : midpoint ℝ C D = O)
    (hPΛ : P ∈ Λ) (hPline : Collinear ℝ {C, D, P}) (hPO : P ≠ O)
    : InnerProductGeometry.angle (A -ᵥ P) (C -ᵥ P) =
      InnerProductGeometry.angle (B -ᵥ P) (D -ᵥ P) := by
  sorry
```

Collinear input format and angle usage syntax errors, easily fixable by an annotator.

Figure 8: Minimal edits needed to fix a broken, but mathematically sound generation, as pointed out by the LLM summary. Reducing human effort by a margin.

ditional benefit: the method can substantially reduce the human effort required for data annotation and theorem formalization. Figure 8 illustrates a representative example where steering produces mathematically sound code that requires only minimal corrections to become a valid formal proof.

In this example, the model generates a Lean theorem about geometric relationships in Euclidean space. The initial generation contains two primary issues: incorrect collinear input formatting and improper angle usage syntax. However, the mathematical reasoning underlying the proof is correct, as confirmed by the model's own natural language summary stating that "gpt-5 provides the cleanest and most mathematically accurate formalization." The required fixes are straightforward syntactic corrections that can be applied by domain experts with minimal effort.

This pattern suggests that activation steering not only improves proof success rates but also generates "near-miss" proofs that are closer to correctness than baseline outputs. Rather than producing completely invalid formal statements, steered models tend to generate mathematically coherent structures with localized syntax errors. This property significantly reduces the annotation burden for creating training datasets, as human experts can focus on lightweight editing rather than complete theorem reconstruction.

The implications for scalable dataset creation are substantial. Traditional approaches to building formal mathematics datasets require expert mathematicians to write complete formalizations from scratch—a time-intensive process that limits dataset scale. Our approach enables a more efficient workflow: models generate candidate formalizations that capture the essential mathematical content, while human experts provide targeted corrections to syntax and edge cases. This collaborative paradigm could accelerate the development of large-scale formal mathematics corpora needed to train more capable theorem-proving systems.

We note that this benefit emerges naturally from our steering methodology rather than being explicitly optimized for. The fact that informal reasoning guidance leads to more structured, correctable outputs suggests that the underlying activation patterns encode not just proof search strategies, but also adherence to formal syntax conventions. This observation warrants further investigation in future work focused specifically on human-AI collaborative formalization workflows.

## A.3 Knowledge Base Learning Prompt

### A.3.1 System Prompt

```
You are a mathematical documentation agent specializing in the Lean 4
Mathlib library. Your job is to explore the Mathlib repository and create
```

```
 a concise, practical documentation summary focused on mathematical
formalization and theorem proving. You have been given access to a yet
unreleased version of this library, which you must go through and pick
out all relevant imports based on the type of problem the user is trying
to solve. The repository contains a comprehensive library of formalized
mathematics for Lean 4.
The repository will have file names and folder names representative of
its content.

Your every response must be a tool call.

The documentation will be used by new lean users, who will use it as a
guide to write all their imports, writing notations and rely solely on it
 to make the correct imports.

WORKFLOW:
1. Use run_bash to explore the repository (ls, cd, cat, grep, find, etc.)
2. Take notes by writing to files in {working_directory}. You are
currently at this directory. Please do not make any changes outside of
this directory, or delete any existing file.
     i.      First read all the given examples, and create a list keywords
, such that each keyword is a concept that appears in any question.
     ii.     Keywords should also include common patterns like how to
express "point lies on line segment", "lines are parallel/perpendicular",
 ratios and divisions of segments.
     iii.    Add these keywords to your notes file, so you can refer to
them for completion later on.
     iv.     Understand the kind of problems the documentation needs to
deal with, and select what goes in accordingly.
3. When you have sufficient information, use final_submit with a complete
 documentation string

EXPLORATION STRATEGY:
- Examine the main mathematical domains asked by the user
- Look for key theorem statements and their dependencies
- Pay attention to naming conventions and mathematical abstractions
- Use the given sample of examples to understand what parts to focus on
- Look for file names, folder names, documentation, examples, source code
 to know their subject
- Focus on user-facing functionality
- Use {working_directory} for any notes (absolute paths since you'll be
changing directories)
- You decide when you have enough information to create the final
documentation

FINAL DOCUMENTATION FORMAT:
Organize your final output into exactly these 4 sections:

## 1. Installation & Import
- How different imports are situated in the mathlib file hierarchy
- Essential import statements for different mathematical domains
- Any setup requirements, like opening some namespace for certain symbols
, literals, notations or declarations.

## 2. Available Namepaces and Symbols
- Group related functionality together
- Since you will be given a field by the user, focus only on that and
related thing you see in the examples
- Important theorem statements in each subdomain
```

```
- Common mathematical objects and their properties
- Exhaustive list of all the functions avaliable for use

## 3. Minimal Usage Example
- Simple theorem statement (with sorry, ignore proofs)
- Basic mathematical definitions
- Make some imports, and open some namespaces and scopes
- All sample codes **must** be complete and well explained, or else it
can confuse the readers on what a complete theorem code looks like
- Do not leave parts of example code as comments
- Give examples for the kind of stuff the reader will be dealing with
when trying to formalize the problem statement
- Lean has difficult type setups, so be sure to explain those with
examples
- Should work out of the box

## 4. Common Pitfalls & Gotchas
- Common mistakes when formalizing mathematics
- Type class resolution issues
- Mathematical notation vs. Lean syntax differences

## 5. Key Files Structure
- An ascii directory tree of all the important/related files and packages

If some concept appears even once in the examples, make sure to cover
that in your documentation. It should be **complete**, don't skip
concepts randomly.
Do not be afraid to make long if it needs to be.

Remember: Your goal is to create a practical cheat sheet that gets
developers productive quickly. It is okay if its long as long as we are
putting relevant information and are correct.
```

### A.3.2 USER MESSAGE

```
Problem Description: I want to understand what all library modules are
available to me for autoformalizing **Set Theory & Combinatorics**
olympiad like problem statements into lean 4. I only care about
autoformalizing the theorem part, so things like tactics and everything
related to solving the problem are unnecessary. Only things relevant to
the theorem statement are useful. I am interested in:
- All the necessary and relevant imports, their correct paths
- How to open the correct namespace or scope to use particular symbols or
 literals in lean
- Examples of using them
- Other things to note
I'll attach some examples of the type of questions I am trying to write
as a lean theorem.

Examples: Samples of the kind of questions whose autoformalization I'll
be doing:
- All the 7-digit numbers containing each of the digits 1, 2, 3, 4, 5, 6,
 7 exactly once, and not divisible by 5, are arranged in the increasing
order. Find the 2000-th number in this list.
- Prove that the number of triples (A, B, C) where A, B, C are subsets of
 {1, 2,  , n} such that ABC = , AB  , BC   is 7 − 26 + 5.
- Let S = {1, 2, . . . , n} and let T be the set of all ordered triples
of subsets of S, say (A1, A2, A3), such that A1  A2  A3 = S. Determine,
```

```
in terms of n, _(A1,A2,A3)T |A1  A2  A3| where |X| denotes the number of
elements in the set X.
- There are 100 countries participating in an olympiad. Suppose \(n\) is
a positive integer such that each of the 100 countries is willing to
communicate in exactly \(n\) languages. If each set of 20 countries can
communicate in at least one common language, and no language is common to
 all 100 countries, what is the minimum possible value of \(n\)?
- A box contains answer 4032 scripts out of which exactly half have odd
number of marks. We choose 2 scripts randomly and, if the scores on both
of them are odd number, we add one mark to one of them, put the script
back in the box and keep the other script outside. If both scripts have
even scores, we put back one of the scripts and keep the other outside.
If there is one script with even score and the other with odd score, we
put back the script with the odd score and keep the other script outside.
 After following this procedure a number of times, there is at least one
script each with odd and even scores. Find, with proof, the number of
scripts with odd scores among the three left.
- The set \( X \) of \( N \) four-digit numbers formed from the digits 1,
 2, 3, 4, 5, 6, 7, 8 satisfies the following condition: for any two
different digits from 1, 2, 3, 4, 5, 6, 7, 8 there exists a number in \(
X \) which contains both of them. Determine the smallest possible value
of \( N \).
- For any natural number n, (n  3), let f(n) denote the number of non-
congruent integer-sided triangles with perimeter n (e.g., f(3) = 1, f(4)
= 0, f(7) = 2). Show that
(a) f(1999) > f(1996);
(b) f(2000) = f(1997).
- Some 46 squares are randomly chosen from a 9 x 9 chess board and are
coloured red. Show that there exists a 2 x 2 block of 4 squares of which
at least three are coloured red.
- A Magician and a Detective play a game. The Magician lays down cards
numbered from 1 to 52 face-down on a table. On each move, the Detective
can point to two cards and inquire if the numbers on them are consecutive
. The Magician replies truthfully. After a finite number of moves the
Detective points to two cards. She wins if the numbers on these two cards
 are consecutive, and loses otherwise. Show if the Detective can
guarantee a win if and only if she is allowed to ask at least 50
questions.
- Let S be a finite set of positive integers. Assume that there are
precisely 2023 ordered pairs (x, y) in S  S so that the product xy is a
perfect square. Prove that one can find at least four distinct elements
in S so that none of their pairwise products is a perfect square.

Please explore the repository and create comprehensive documentation
following the 4-section format. Start by exploring the current directory
structure to understand what you're working with.
Your working directory is {working_directory}. Please refrain from doing
anything outside of this directory, or deleting any of its content. You
may create your notes file here if you want to.
```

### A.3.3 ITERATIVE REFINEMENT PROMPT

We omit the solution part for problems without a solution.

```
You are an expert at writing Lean code. Your task is to convert a
naturallanguage informal question into a Lean 4 formalized statement only
 (no proofs). Work entirely from first principles and axiomsdo **not**
assume or derive the proof.
```

```
**Output format** (and nothing else):
```lean
...
```

---
Problem {problem['id']}:
{problem['informal_question']}

Solution (for context  incorporate the necessary details into the theorem
 statement, but do **not** include a proof):
{solution}
```

