# OpenReview forum: "IndiMathBench: Autoformalizing Mathematical Reasoning Problems with a Human Touch"
_ICLR.cc/2026/Conference — ICLR 2026 Conference Desk Rejected Submission_

### Official Review · Reviewer_empU · 2025-10-29

**Soundness:** 2
**Presentation:** 2
**Contribution:** 1
**Rating:** 2
**Confidence:** 4

**Summary:**

This research introduces IndiMathBench, a new benchmark for automatic theorem proving. It consists of 312 mathematical problems sourced from the Indian Mathematical Olympiad, each paired with a human-verified formalization in the Lean 4 theorem prover.

The benchmark was created using a novel AI-powered human-assisted pipeline. In this process, researchers first utilized an ensemble of LLMs with a category-based retrieval and self-correction loop to generate initial formalizations. Human experts then validated and corrected these AI-generated proposals through an interactive dashboard. Several ablation studies and comparative studies are designed to demonstrate the effectiveness of this pipeline. To facilitate collaboration between humans and AI for Lean annotations, the researchers also developed a Visual Studio Code extension aimed at improving the efficiency of the formalization process.

**Strengths:**

1. The formalization of the problems presented in the paper is of good quality, especially the clever formalization of the three integer roots in the problem inmo_2017_2 in appendix A.
2. The pipeline of this paper has indeed improved the annotation efficiency by 2-3 times that of human experts, which is a certain improvement.
3. It is original and has engineering value. It transforms an AI-assisted formalization framework into a user-friendly VSCode extension, which may be useful for the development of the Lean community.

**Weaknesses:**

The two main contributions of the paper, the **formal benchmark** itself and the **autoformalization pipeline**, both have certain weaknesses.

Regarding the formal benchmark itself,

1. The benchmark is not substantially different from the existing IMO benchmark, and its coverage remains almost identical. Other formal benchmarks at the high school competition level, such as miniF2F [1] and Fimo [2], already exist.
2. Concerning the issue of data contamination with MiniF2F and IMO, I believe that all public competition problems have comparable likelihood of having online solutions, while formalized solutions are generally unavailable. In this regard, IndiMathBench does not demonstrate a clear advantage. Additional analysis or empirical evidence is needed to support the claim that IndiMathBench can effectively mitigate data contamination.
3. For the evaluation of this benchmark, I recommend conducting further experiments. First, for the formal theorem proving task, it would be more convincing to evaluate state-of-the-art theorem provers, such as Deepseek-prover-v2 [3], Kimina-prover [4], and Goedel-prover-v2 [5], with increased sampling budgets to obtain more robust results. Second, the experiments in Section 5.2 only involve syntax checking rather than semantic checking. Incorporating semantic checking would enhance the soundness of the evaluation.

Regarding the autoformalization pipeline:

1. Each component of the pipeline has stronger existing alternatives. For example, iterative refinement and LLM judges have already been adopted in recent works like Kimina-Autoformalizer [4]. The category-based retrieval method relies on static documentation, which is not sufficiently comprehensive. In the “Knowledge base for Mathlib” section of Figure 2, the group-related results shown are irrelevant to the intended topic.
2. This work should also include a comparison with existing autoformalizers (e.g., Kimina-Autoformalizer, Goedel-Formalizer-V2 [5], and Herald [6]) on InMO and RMO benchmarks to better contextualize its contribution.
3. In the ablation study corresponding to Table 3, the KB and FB components are evaluated together. Conducting additional experiments to separately measure the individual contributions of KB and FB would yield a clearer understanding of their respective effects.

[1] Zheng, K., Han, J. M., & Polu, S. (2021). Minif2f: a cross-system benchmark for formal olympiad-level mathematics. arXiv preprint arXiv:2109.00110.

[2] Liu, C., Shen, J., Xin, H., Liu, Z., Yuan, Y., Wang, H., ... & Liu, Q. (2023). Fimo: A challenge formal dataset for automated theorem proving. arXiv preprint arXiv:2309.04295.

[3] Ren, Z. Z., Shao, Z., Song, J., Xin, H., Wang, H., Zhao, W., ... & Ruan, C. (2025). Deepseek-prover-v2: Advancing formal mathematical reasoning via reinforcement learning for subgoal decomposition. arXiv preprint arXiv:2504.21801.

[4] Wang, H., Unsal, M., Lin, X., Baksys, M., Liu, J., Santos, M. D., ... & Li, J. (2025). Kimina-prover preview: Towards large formal reasoning models with reinforcement learning. arXiv preprint arXiv:2504.11354.

[5] Lin, Y., Tang, S., Lyu, B., Yang, Z., Chung, J. H., Zhao, H., ... & Jin, C. (2025). Goedel-prover-v2: Scaling formal theorem proving with scaffolded data synthesis and self-correction. arXiv preprint arXiv:2508.03613.

[6] Gao, G., Wang, Y., Jiang, J., Gao, Q., Qin, Z., Xu, T., & Dong, B. (2024). Herald: A natural language annotated lean 4 dataset. arXiv preprint arXiv:2410.10878.

**Questions:**

1. Why does the Masked Candidates method run at roughly twice the time cost compared to the Full system?
2. How many human expert annotators contributed to the benchmark, and what are their academic or professional backgrounds?
3. How well does the proposed pipeline generalize to non-competition problems, such as standard undergraduate-level mathematics problems?

---

> ### Author Response · Authors · 2025-11-25
> **Responses to Reviewer empU Part 1/5**
>
> First, we would like to thank the reviewer for their detailed comments on our work. We sincerely value them and take them very seriously. Please feel free to ask any questions again if any part remains unclear.
>
> ### **W1: Benchmark Difference**
>
> > The benchmark is not substantially different from the existing IMO benchmark, and its coverage remains almost identical.
>
> We appreciate the reviewer’s concern and clarify that IndiMathBench is not intended as a general replacement for existing Olympiad datasets, but as a complementary benchmark that fills well-documented gaps in the current Lean 4 ecosystem. It is similar in spirit to existing Olympiad-level Lean 4 benchmarks (e.g., miniF2F, PutnamBench), but its contribution lies in the areas where current benchmarks remain structurally sparse. In particular, Olympiad-level geometry, combinatorics, and set-theoretic problems are underrepresented across all existing datasets, despite being the categories where both ATPs and LLM-guided provers struggle the most. Recent efforts such as LeanGeo [1] and CombiBench [2] were motivated by this same gap, indicating that it is a well-recognized limitation in the formalization ecosystem.
>
> To make this discrepancy clear, we include the following comparison, aggregating counts directly from the cited benchmark papers:
>
> | Olympiad Benchmark | Geometry Problem Count | Notes/Sources                                                |
> | ------------------ | ---------------------- | ------------------------------------------------------------ |
> | PutnamBench        | 68                     | Putnam problems; few synthetic-geometry constructions        |
> | IMO                | ~45                    | MINIF2F, Seed-Prover [3], IMO-Bench                          |
> | LeanGeo            | 19                     | Out of 122 geometry problems, only 19 are Olympiad-level [1] |
> | FIMO               | 0                      | Focuses on non-geometry tasks [4]                            |
> | **Total**          | **~132**               | Approx. combined Olympiad-level geometry coverage            |
> | **IndiMathBench**  | **98**                 | IndiMathBench adds a significant amount of Geometry problems |
>
> Olympiad-level geometry remains especially hard to formalize in Lean, with both ATPs and LLM-guided provers performing poorly. This difficulty has been repeatedly documented: geometry benchmarks such as Seed-Geo [3] and LeanGeo [1] highlight that current Lean 4 libraries lack native support for many synthetic constructions, auxiliary objects, and transformation rules required in Olympiad solutions; similarly, LLMs perform comparatively weakly on geometry-intensive tasks (AlphaGeometry [5] and Seed-Geo [3]). The scarcity has directly motivated the creation of dedicated datasets like LeanGeo, which introduced 122 new problems and a library for Olympiad style Geometry. IndiMathBench extends this space with a large set of formalized high-difficulty geometry problems.
>
> A similar gap exists in Olympiad-level combinatorics. Lean datasets contain very little content at this difficulty level, prompting dedicated efforts such as CombiBench [2] (100 problems). IndiMathBench adds 45 new combinatorics and set-theory problems involving multi-step casework, extremal arguments, and counting techniques that are currently underrepresented.
>
> Indian Olympiad problems also feature algebraic and number-theoretic styles not common in Western competitions. For these reasons, **we respectfully clarify that IndiMathBench is not identical in coverage to existing IMO-based datasets**; its formalized coverage differs meaningfully by adding substantial formalized content in categories, especially Olympiad-level geometry, combinatorics, and non-Western algebraic styles, that current benchmarks include only sparsely or not at all.
>
>
> [1] Song et al. LeanGeo: Formalizing Competitional Geometry Problems in Lean (No. arXiv:2508.14644). arXiv. [https://arxiv.org/abs/2508.14644](https://arxiv.org/abs/2508.14644)
> [2] Liu et al. CombiBench: Benchmarking LLM Capability for Combinatorial Mathematics (No. arXiv:2505.03171). arXiv. [https://arxiv.org/abs/2505.03171](https://arxiv.org/abs/2505.03171)
> [3] Chen et al. Seed-Prover: Deep and Broad Reasoning for Automated Theorem Proving (No. arXiv:2507.23726). arXiv. [https://arxiv.org/abs/2507.23726](https://arxiv.org/abs/2507.23726)
> [4] Liu et al. FIMO: A Challenge Formal Dataset for Automated Theorem Proving (No. arXiv:2309.04295). arXiv. [https://github.com/liuchengwucn/FIMO](https://github.com/liuchengwucn/FIMO)
> [5] Chervonyi et al. Gold-Medalist Performance in Solving Olympiad Geometry with AlphaGeometry2 (No. arXiv:2502.03544). arXiv. https://arxiv.org/abs/2502.03544

---

> > ### Author Response · Authors · 2025-11-25
> > **Responses to Reviewer empU Part 2/5**
> >
> > ### **W2: Data Contamination**
> >
> > We appreciate the reviewer’s concern and agree that public competition problems can in principle appear online. To be clear, the contamination we are referring to is not a matter of a system utilizing informal solutions found online, but of the LLM itself having ingested the problem text into its parametric memory during pretraining.
> >
> > From this perspective, contamination risk is not determined by whether a problem is “somewhere on the internet,” but by whether the problem exists in curated, machine-readable, repeatedly mirrored forms that are commonly ingested into pretraining corpora. This distinction is emphasized in prior LLM contamination analyses (e.g., Roberts et al., 2023 [6]; Bubeck et al., 2023 [7]), which document that widely replicated and structured benchmarks have significantly higher contamination likelihood than fragmented, low-resource sources.
> >
> > In this context, datasets such as miniF2F, FIMO, and PutnamBench are all explicitly released as clean, canonical, machine-readable datasets (JSON/YAML/Lean files). They are widely mirrored across GitHub, HuggingFace, and academic repositories, ensuring their high visibility to crawler heuristics used by model trainers.
> >
> > IndiMathBench differs in two concrete ways:
> > 1. The problems were not previously available as a curated dataset—they existed only as scattered PDFs, scanned booklets, or low-resource competition archives, which are far less likely to be included or retained during corpus deduplication.
> > 2. No prior formalized versions of these problems existed, whereas MiniF2F and FIMO explicitly provide cleaned, tokenized, machine-readable representations that make contamination more direct.
> >
> > We do not claim that IndiMathBench completely eliminates contamination risk; rather, it has a meaningfully lower contamination profile compared to benchmarks that have served for several years in standardized, high-quality forms. This is the same practical strategy used in other evaluation suites such as LiveCodeBench [8], which explicitly rely on “post-cutoff” data to measure model generalization. As with all such benchmarks, contamination risk naturally increases over time, and periodic updates become necessary. Our point is not that this solves contamination, but that it provides a pragmatic, lower-contamination evaluation today.
> >
> > [6] Roberts et al. Data Contamination Through the Lens of Time (No. arXiv:2310.10628). arXiv. https://arxiv.org/abs/2310.10628
> > [7] Bubeck et al. Sparks of Artificial General Intelligence: Early Experiments with GPT-4 (No. arXiv:2303.12712). arXiv. https://arxiv.org/abs/2303.12712
> > [8] Jain et al. LiveCodeBench: Holistic and Contamination Free Evaluation of Large Language Models for Code (No. arXiv:2403.07974). arXiv. https://arxiv.org/abs/2403.07974

---

> ### Author Response · Authors · 2025-11-25
> **Responses to Reviewer empU Part 3/5**
>
> ### **W3 and W5: Expanded Model Coverage**
>
> Thank you for this recommendation! Over the rebuttal phase, we extended our ATP evaluation from general-purpose LLMs, to SOTA fine-tuned theorem-proving models, including DeepSeekProverV2 7B and GodelProverV2 8B as recommended by you, under exactly the same scaffold.
>
> **Introduction of a Complete ATP Scaffold**
> We also implemented a more complete ATP system that integrates multi-step reasoning and verification feedback from the Lean compiler. The new setup uses a ReAct-style iterative interaction loop over 10 refinement turns, where the model is allowed to reflect on compiler errors and generate corrected Lean code. This design mirrors the structure of systems such as DeepSeek-Prover, while remaining computationally feasible within the constraints of our evaluation. We run this at pass@1 because high-pass@k scaffolds (with k in the hundreds or thousands) are prohibitively expensive for our setting.
>
> Theorem Proving resolution rates across IndiMathBench (IMB), and PutnamBench (PB):
>
> | Model               | Single Turn IMB | 10 Turns IMB | 10 Turns PB |
> | ------------------- | --------------- | ------------ | ----------- |
> | GPT-4.1             | 0/312           | -            | -           |
> | o3 (medium)         | 1/312           | -            | -           |
> | Claude Sonnet 4     | 1/312           | 4/312        | -           |
> | Gemini 2.5 Pro      | 0/312           | 12/312       | -           |
> | GPT-5               | 1/312           | 36/312       | 28/660      |
> | DeepSeekProverV2 7B | 3/312           | 24/312       | -           |
> | GodelProverV2 8B    | 7/312           | 36/312       | -           |
>
> **Regarding the Results in Section 5.2**
> The semantic analysis and results of Autoformalization capabilities are covered in Section 5.1. Where we also extend the evaluation to two new open source autoformalization SLMs under your recommendation.
>
> The results in Section 5.2 are meant to indicate how helpful step 2 “Iterative Refinement with Error Feedback” is in our Autoformalization pipeline. Since internally in step 2 we stop the interactions for theorem generation once we reach a valid compilable lean snippet.
>
> **Comparison with existing Autoformalizers**
> We compare the existing Autoformalizers recommended by you (Kimina-Autoformalizer, Goedel-Formalizer-V2) on the same setup (iterative refinement, using a knowledge base), under the same metrics. The results are especially interesting that Godel formalizer is able to score as well as Claude Opus 4 in BEq. We believe this is the first extensive work comparing such a large variety of models on Autoformalization and ATP.
>
> | Model                    | BEq | GTED Mean | #GTED > 0.9 | Compile Success |
> | ------------------------ | --- | --------- | ----------- | --------------- |
> | Claude Opus 4            | 54  | 0.51      | 138         | 243             |
> | Claude Sonnet 4          | 51  | 0.42      | 103         | 215             |
> | Gemini 2.5 Pro           | 47  | 0.24      | 58          | 151             |
> | GPT-5                    | 38  | 0.48      | 124         | 235             |
> | o3 (high)                | 32  | 0.46      | 119         | 205             |
> | GPT-4.1                  | 29  | 0.39      | 90          | 120             |
> | Kimina-Autoformalizer 7B | 17  | 0.19      | 38          | 101             |
> | Godel-Formalizer-V2 7B   | 52  | 0.47      | 119         | 222             |
>
> We will include these results and its detailed qualitative analysis in the revised paper.

---

> > ### Author Response · Authors · 2025-11-25
> > **Responses to Reviewer empU Part 4/5**
> >
> > ### **W4: Pipeline not being SOTA**
> >
> > Thank you for your critique regarding the pipeline components. We would like to clarify the specific design choices and their impact on our primary contribution: the IndiMathBench dataset and the **reduction of human annotation effort**.
> >
> > 1. Comparison to Existing Pipelines: While we acknowledge that iterative refinement and LLM judges are used in multiple works, and are very common, our contribution lies in systematically adapting these specifically for Human-AI Collaborative Annotation rather than fully autonomous proving.
> > 	- Different Objective: Unlike standard autoformalizers optimized for solving, our pipeline is optimized to produce "human-fixable" code. As shown in Figure 4, our system reduces human annotation time by 3.5x (from 14 mins to 4 mins).
> > 	- Ensemble Efficacy: Our specific combination of multi-model generation (Ensemble) + Compiler Feedback achieves a 95.3% compilation coverage success rate (Section 4, Step 2). This high reliability is critical for the human-in-the-loop workflow we propose, which is distinct from the objectives of pure autoformalization papers.
> > 	- Other works have used such a LLM generation with human annotation pipeline before (FIMO [4], FormalMATH [9]), but as per our knowledge, we are the first to introduce a systematic UI that aggregates multi-model generations, summarizes "best parts" via an ensemble, and provides compiler feedback to the human.
> > 	- Our approach is analogous to a radiologist using AI systems to highlight potential issues for efficient diagnosis, rather than an AI attempting fully autonomous diagnosis. We believe this "human-in-the-loop" focus is critical for creating high-quality, reliable datasets.
> >
> > 2. Efficacy of Category-Based Retrieval: The reviewer suggests static documentation is insufficient. However, our empirical results contradict this.
> > 	- In Table 3, we show that adding our Documentation (KB) and Feedback (FB) loops improves success rates drastically (e.g., GPT-4.1 improves from 58 to 132 successes; Claude Opus 4 improves from 17 to 310).
> > 	- Lean 4 formalization often fails due to missing/hallucinated imports or syntax conventions rather than missing specific lemmas. Our static "cheat sheet" (Appendix A.3.1) ensures the correct environment is established, that the model has a better idea of its “world” (mathlib in this case).
> > 	- Olympiad problems generally rely on foundational theorems rather than niche, problem-specific constructions. Our ~200-line generated Knowledge Base covers these major topics effectively. We argue that granular retrieval is overkill.
> > 	- We include a snippet of a sample knowledge base document generation for reference:
> >
> >
> > ```
> > Below is Mathlib library documentation on some relevant packages that you might need to use for your task. This is not an exhaustive list of Mathlib features, just some of them.
> >
> > # Mathlib Lean 4 Reference (Algebra, Analysis, Geometry)
> >
> > ## Overview
> >
> > * **General structure:** Mathlib is organized by topic in the `Mathlib` namespace, e.g. `Mathlib.Algebra`, `Mathlib.Analysis`, `Mathlib.Geometry`. ...
> >
> > ## Number Systems
> > * ℕ: `Mathlib.Data.Nat.Basic`
> > * ℤ: `Mathlib.Data.Int.Basic`
> > * ℚ: `Mathlib.Data.Rat.Basic`
> > * ℝ: `Mathlib.Data.Real.Basic` (as Cauchy sequences)
> > * ℂ: `Mathlib.Data.Complex.Basic`
> >
> > ...
> > ## Analysis (Topology and Calculus)
> > * **Topology and Metric Spaces:**
> >   * Modules: `Mathlib.Topology.MetricSpace.Basic`, `Mathlib.Topology.Continuity`
> >   * Includes: Filters, continuity, compactness, Bolzano-Weierstrass, Heine-Borel.
> >
> > * **Uniform Spaces and Completeness:**
> >   * Modules: `Mathlib.Topology.UniformSpace.Basic`
> >   * Concepts: Cauchy filters, uniform continuity.
> > ...
> > ## Special Functions
> > * **Trigonometric Functions:**
> >   * Module: `Mathlib.Analysis.SpecialFunctions.Trigonometric.Basic`
> >   * Functions: `Real.sin`, `Real.cos`, `Real.pi`
> >   * Includes: Identities, periodicity, special values.
> > ...
> > ```
> >
> > Hopefully this gives a better picture of how one static knowledge base document is enough for an entire category of problems.
> >
> > 3. We apologize if Figure 2 gave the impression that the Knowledge Base is limited to the displayed lines. It is only the first few lines of a rather long knowledge base covering a variety of topics and imports. The problem is only one randomly selected problem. The document is supposed to cover all the topics seen in the 25% sample the retrieval agent is given, which we observe that it usually does. We will update the figure to display a more representative section of the retrieved context and clarify this in the caption. Thank you for pointing these out!
> >
> > [9] Yu et al. FormalMATH: Benchmarking Formal Mathematical Reasoning of Large Language Models (No. arXiv:2505.02735). arXiv. [https://arxiv.org/abs/2505.02735](https://arxiv.org/abs/2505.02735)

---

> > > ### Author Response · Authors · 2025-11-25
> > > **Responses to Reviewer empU Part 5/5**
> > >
> > > ### **W6: KB and FB Ablation**
> > >
> > > Thank you for this recommendation! We were able to perform some experiments during the rebuttal for the ablation of FB individually. We will report these results, and the analysis of improvements seen from FB to KB+FB in detail in the revised paper. Overall we see an improvement across the board, relatively much more in case of Gemini 2.5 Pro. Most of the gains come from the iterative feedback loops.
> > >
> > > | Model             | ZS (%) | FB (%) | KB+FB (%) |
> > > | ----------------- | ------ | ------ | --------- |
> > > | Claude Sonnet 3.7 | 8.2    | 41.9   | 49.3      |
> > > | o3 (high)         | 22.1   | 53.9   | 65.7      |
> > > | Gemini 2.5 Pro    | 3.8    | 27.4   | 48.4      |
> > > | GPT-4.1           | 13.9   | 30.3   | 38.5      |
> > > | GPT-5             | 30.5   | -      | 75.3      |
> > > | Claude Opus 4     | 4.1    | -      | 77.9      |
> > >
> > > ### **Q1: Masked candidate setting results**
> > > The annotator in the study states mainly two reasons for the 2x speedup when switching from masked to full setting:
> > > - Lack of model summary: The Full System includes an LLM-generated summary that ranks the models and explicitly points out what the best generation got wrong and where to find the correct formalization for that specific part in other generations. This helps the annotator “speed pick” the right parts to replace from the wrong ones. The masked setting forced the annotator to find these correct segments manually.
> > > - Cognitive Load: The annotator noted that the summary significantly reduced the effort by directing why something went wrong when the top candidate failed, whereas, in the Masked Candidates method, understanding the logic took some time.
> > >
> > > Thank you for this question. We will incorporate these insightful answers into the revised paper as well.
> > >
> > > ### **Q2: How many human expert annotators contributed to the benchmark, and what are their academic or professional backgrounds?**
> > >
> > > The formalization was done over the course of a month by two annotators from an industrial research lab with 1–2 years of active experience in using Lean for formal proof writing. Every annotation was independently double checked by the other annotator. In case of difficult problems, or disagreements over the style or correctness of the annotations, the annotators discussed the theorem and came to a final answer. A common code style was followed to make the entire problem be represented within a single theorem statement as possible, and use similar constructions for similar concepts across the benchmark.
> > >
> > > We will include these details in the revised version.
> > >
> > > ### **Q3: How well does the proposed pipeline generalize to non-competition problems, such as standard undergraduate-level mathematics problems?**
> > >
> > > While evaluated on Olympiad math, our pipeline is methodologically domain-agnostic.
> > > 1. Undergraduate Math: The Category-Based Retrieval adapts to advanced topics (e.g., Topology) simply by pointing the documentation agent to those specific libraries. No other parts need any change.
> > > 2. Beyond Math: As noted in Section 4.2, the dashboard and feedback loops are designed for general "informal to formal" tasks. The workflow: generating candidates, verifying syntax via compiler, and summarizing differences, is directly extensible to domains like NL-to-SQL or NL-to-Spreadsheet-Formula, where enforcing syntactic correctness and human disambiguation are critical.
> > > 3. Tracking Human edits: The dashboard is also usable to track human edits to generate better data for RL feedback for models.
> > >
> > > ### **S3: Open Release Commitment**
> > > Thank you for the encouraging feedback. A central goal of this work is to let the community see firsthand how human–AI mixed systems can meaningfully accelerate human labor, annotation and analysis tasks in our case. Hence, openness has been a target of this project from the outset, because the intended impact of our human–AI interface depends on the community’s ability to examine, reproduce, and extend the work. Since the project sits within a corporate environment, open sourcing these assets and the benchmark has required navigating a multi-team approval pipeline that has spanned several months. We have invested substantial effort throughout this process to secure a responsible, unrestricted release. We remain fully committed to completing it so the community can directly benefit from, scrutinize, and build upon our contributions.

---

### Official Review · Reviewer_f4fs · 2025-10-29

**Soundness:** 3
**Presentation:** 2
**Contribution:** 2
**Rating:** 6
**Confidence:** 4

**Summary:**

This paper presents IndiMathBench, an automated theorem proving benchmark dataset that has been verified by human experts. The dataset is constructed using a formalization approach where AI plays the primary role and humans assist, with problem statements sourced from the Indian Mathematics Olympics, comprising a total of 312 problems. The authors propose an autoformalization pipeline featuring a category-based retrieval mechanism and a self-debug loop integrated with Lean, where multiple LLMs generate several formalization candidates. Additionally, the paper implements a VS Code Extension, and dashboard designed to accelerate the data annotation process. The study also provides a comparative analysis of the capabilities of several state-of-the-art models in formalizing problems from the Indian Mathematics Olympics.

**Strengths:**

1. IndiMathBench covers problem types such as set theory, combinatorics, and geometry that are not typically included in existing ATP benchmark datasets (like miniF2F). This diversity enhances the practical value of the dataset as a comprehensive ATP evaluation resource.

2. The proposed VS Code dashboard holds considerable potential for accelerating data annotation in this field. As shown in Figure 4 of the paper, the pipeline seems to significantly reduce the time required to annotate formal mathematical statements. The idea of allowing human annotators to reuse parts across multiple formalization candidates to improve labeling efficiency is both interesting and sensible.

3. The analysis of formalization capabilities across various models, including state-of-the-art ones such as Claude Sonnet 4, Claude Opus 4, and OpenAI o3 High, provides valuable insights for selecting LLMs in future autoformalization work.

4. The use of the relatively recent Lean 4.22 version ensures that the dataset remains relevant and aligned with current formalization tool environments.

**Weaknesses:**

1. The proposed autoformalization pipeline is not truly novel; rather, it largely applies existing techniques to the Indian Mathematics Olympics dataset with some engineering optimizations. Specifically, the idea of a self-debug loop using error messages from the Lean compiler has appeared in prior works [1]; the concept of human feedback-assisted adjustment has been explored in earlier work [2]; and the use of retrieval from mathlib to enhance autoformalization capability is also present in previous research [3].

2. The AUTOMATED THEOREM PROVING EVALUATION section relies on general-purpose LLMs for evaluation and lacks benchmark results from domain-specific automated theorem provers. This choice is unusual and not best practice within the field. Evaluating with general-purpose frontier LLMs alone typically yields poor results in theorem proving; at minimum, inclusion of specialized open-weight provers such as Deepseek Prover V2 or Goedel Prover v2 is needed. Given that Deepseek-Prover is cited in the paper (Line 100), the absence of its evaluation here is a notable omission. As a result, the experimental data in Table 4 are insufficient to convincingly demonstrate the dataset’s difficulty.

3. The paper applies a static context retrieval approach per problem category, rather than generating retrievals tailored to each individual query as is common in typical RAG methods. Lines 239 (“There is one static context retrieved for each category of problems”) and Line 256 acknowledge the static nature of retrieval. Since each category may encompass numerous concepts (especially given the dataset is divided into only four broad categories; for example, the algebra category likely covers a wide range of prerequisite knowledge), employing a single static context per category potentially limits the pipeline’s generalizability and may restrict successful formalization to contents covered by the static context.

4. The writing contains several minor errors that, although not obstructing readability, give the impression of hastiness. The issues noted include:

   - Inconsistent spelling of “Lean4” and “Lean 4”; these should be standardized (e.g., Lines 18, 53, 58, and Figure 1 caption).
   - Inconsistent casing of “MINIF2F” and “miniF2F” across the manuscript (e.g., Lines 75, 100, 162, 184).
   - Possible typo “Wxe” on Line 53.
   - “Visual Studio” on Line 60 should be corrected to “Visual Studio Code.”
   - Missing appendix references on Line 263.
   - Missing figure number citation on Line 295.
   - Caption font size for Figure 5 is too small and difficult to read.
   - Table 2 is placed below Tables 3 and 4, which is somewhat confusing.

   [1] Lu, J., Wan, Y., Liu, Z., Huang, Y., Xiong, J., Liu, C., Shen, J., Jin, H., Zhang, J., Wang, H., Yang, Z., Tang, J., & Guo, Z. (2024). *Process-Driven Autoformalization in Lean 4* (No. arXiv:2406.01940). arXiv. https://doi.org/10.48550/arXiv.2406.01940

   [2] Liu, C., Shen, J., Xin, H., Liu, Z., Yuan, Y., Wang, H., Ju, W., Zheng, C., Yin, Y., Li, L., Zhang, M., & Liu, Q. (2023). *FIMO: A Challenge Formal Dataset for Automated Theorem Proving* (No. arXiv:2309.04295). arXiv. https://doi.org/10.48550/arXiv.2309.04295

   [3] Liu, Q., Zheng, X., Lu, X., Cao, Q., & Yan, J. (2024, October 4). *Rethinking and Improving Autoformalization: Towards a Faithful Metric and a Dependency Retrieval-based Approach*. The Thirteenth International Conference on Learning Representations. https://openreview.net/forum?id=hUb2At2DsQ

**Questions:**

1. As noted in Weakness 2, my primary concern is why the work does not include evaluations using domain-specific automated theorem provers, given that this dataset is intended as an automated theorem proving evaluation set and that utilizing such provers is standard practice in the ATP field. Could the authors clarify the reasoning behind this omission?

2. In constructing the autoformalization pipeline, the paper employs an LLM-as-a-judge approach to assess semantic alignment. However, when comparing the autoformalization capabilities of various frontier models later on, the evaluation does not seem to leverage the LLM-as-a-judge method (BEq and GTED only). Considering that LLM-as-a-judge has gained widespread adoption in the autoformalization domain [4], I am curious about how these models would perform if evaluated under this criterion. For example, which models tend to rank highest in zero-shot settings? What are the pass rates for each model? Additional details along these lines would be highly valuable.

3. Line 396 mentions that Claude Sonnet 4 was used to compute BEq instead of following the original BEq implementation, but the paper does not explain the reason for this choice. Could the authors elaborate on why the model was replaced and what motivated this decision?

[4] Ying, H., Wu, Z., Geng, Y., Wang, J., Lin, D., & Chen, K. (2024). Lean workbook: A large-scale lean problem set formalized from natural language math problems. *Advances in Neural Information Processing Systems*, *37*, 105848-105863.

---

> ### Author Response · Authors · 2025-11-25
> **Responses to Reviewer f4fs Part 1/3**
>
> First, we would like to thank the reviewer for their detailed comments on our work. We sincerely value them and take them very seriously. Please feel free to ask any questions again if any part remains unclear.
> ### **W1: Pipeline novelty**
>
> We appreciate the reviewer’s insightful observation. We agree that components like iterative refinement and retrieval are established techniques in the field. But, we would like to clarify that our goal was to engineer a specific integration of these methods to solve a different problem: **reduction of human annotation effort**, and systematically demonstrate it by creating a decent benchmark by using it ourselves.
>
> 1. **Adaptation for Human-AI Collaboration:** While we utilize established techniques like iterative refinement and LLM judges, our specific contribution is the adaptation of these tools into a human-centric workflow rather than an autonomous one.
> 	- Different Objective: Unlike standard autoformalizers optimized for solving, our pipeline is optimized to produce "human-fixable" code. As shown in Figure 4, our system reduces human annotation time by 3.5x (from 14 mins to 4 mins).
> 	- Other works have used such a LLM generation with human annotation pipeline before (FIMO, FormalMATH), but as per our knowledge, we are the first to introduce a systematic UI that aggregates multi-model generations, summarizes "best parts" via an ensemble, and provides compiler feedback to the human. The special focus is maximizing annotator efficiency is unique to our system.
> 	- Our approach is analogous to a radiologist using AI systems to highlight potential issues for efficient diagnosis, rather than an AI attempting fully autonomous diagnosis. We believe this "human-in-the-loop" focus is critical for creating high-quality, reliable datasets in domains where fully automated solutions are not yet 100% reliable.
> 2. Existing Retrieval-Based methods: Regarding retrieval, the reviewer is correct that utilizing Mathlib for retrieval is a standard practice. However, we found that even using a simple and cheap retrieval for every category gives significant gains. Instead of retrieving specific lemmas, we implemented a Category-Based Retrieval system that generates a static "cheat sheet" (Knowledge Base) covering the imports and notations for an entire domain (e.g., Algebra or Geometry).
>
> ### **W2 and Q1: Expanded Model Coverage**
>
> Thank you for this recommendation! Over the rebuttal phase, we extended our ATP evaluation from general-purpose LLMs, to SOTA fine-tuned theorem-proving models, including DeepSeekProverV2 7B and GodelProverV2 8B, as recommended by you, under exactly the same scaffold.
>
> **A Complete ATP Scaffold.**
> We also implemented a more complete ATP system that integrates multi-step reasoning and verification feedback from the Lean compiler. The new setup uses a ReAct-style iterative interaction loop over 10 refinement turns, where the model is allowed to reflect on compiler errors and generate corrected Lean code. This design mirrors the structure of systems such as DeepSeek-Prover, while remaining computationally feasible within the constraints of our evaluation. We run this at pass@1 because high-pass@k scaffolds (with k in the hundreds or thousands) are prohibitively expensive for our setting.
>
> Our overall Theorem Proving resolution rates across IndiMathBench (IMB), and PutnamBench (PB):
>
> | Model               | Single Turn IMB | 10 Turns IMB | 10 Turns PB |
> | ------------------- | --------------- | ------------ | ----------- |
> | GPT-4.1             | 0/312           | -            | -           |
> | o3 (medium)         | 1/312           | -            | -           |
> | Claude Sonnet 4     | 1/312           | 4/312        | -           |
> | Gemini 2.5 Pro      | 0/312           | 12/312       | -           |
> | GPT-5               | 1/312           | 36/312       | 28/660      |
> | DeepSeekProverV2 7B | 3/312           | 24/312       | -           |
> | GodelProverV2 8B    | 7/312           | 36/312       | -           |
>
> >Could the authors clarify the reasoning behind this omission?
>
> During the initial experiments, limited GPU availability forced us to rely solely on API-based commercial models. After the reviewer’s request, we secured additional compute during the rebuttal phase and have now added evaluations on state-of-the-art open-weight models for completeness and reproducibility. The experiments now reflect SOTA models, on decently well performing ATP scaffolds.
>
> Remarkably, Godel-Prover-v2 achieved performance levels comparable to GPT-5 under identical experimental settings. External validation and decent scores on PutnamBench also clarify the efficacy of our scaffold. It definitely goes to show how the latest large reasoning models are catching up to the performance of domain specific fine tuned ones.
>
> We will add these new results and their qualitative analysis to the revised paper.

---

> > ### Author Response · Authors · 2025-11-25
> > **Responses to Reviewer f4fs Part 2/3**
> >
> > ### **W3:  Efficacy of Category-Based Retrieval**
> >
> > The reviewer suggests static documentation is insufficient. However, our empirical results contradict this.
> >
> > - In Table 3, we show that adding our Documentation (KB) and Feedback (FB) loops improves success rates drastically (e.g., GPT-4.1 improves from 58 to 132 successes; Claude Opus improves from 17 to 310).
> > - Lean 4 formalization often fails due to missing/hallucinated imports or syntax conventions rather than missing specific lemmas. The retrieval only needs to provide the overall skeleton of the relevant mathlib files/directories. Our static "cheat sheet" (Appendix A.3.1) ensures the correct environment is established, that the model has a better idea of its “world” (mathlib in this case).
> > - Olympiad problems generally rely on foundational theorems rather than niche, problem-specific constructions. Our ~200-line generated Knowledge Base covers these major topics effectively. We argue that granular retrieval is overkill, adds cost and latency without guaranteed returns for this specific domain. Cost increases from 4\*50 turns to 312\*50 turns.
> > - We include a snippet of a sample knowledge base document generation for reference:
> >
> > ```
> > Below is Mathlib library documentation on some relevant packages that you might need to use for your task. This is not an exhaustive list of Mathlib features, just some of them.
> >
> > # Mathlib Lean 4 Reference (Algebra, Analysis, Geometry)
> >
> > ## Overview
> >
> > * **General structure:** Mathlib is organized by topic in the `Mathlib` namespace, e.g. `Mathlib.Algebra`, `Mathlib.Analysis`, `Mathlib.Geometry`. ...
> >
> > ## Number Systems
> > * ℕ: `Mathlib.Data.Nat.Basic`
> > * ℤ: `Mathlib.Data.Int.Basic`
> > * ℚ: `Mathlib.Data.Rat.Basic`
> > * ℝ: `Mathlib.Data.Real.Basic` (as Cauchy sequences)
> > * ℂ: `Mathlib.Data.Complex.Basic`
> >
> > ...
> > ## Analysis (Topology and Calculus)
> > * **Topology and Metric Spaces:**
> >   * Modules: `Mathlib.Topology.MetricSpace.Basic`, `Mathlib.Topology.Continuity`
> >   * Includes: Filters, continuity, compactness, Bolzano-Weierstrass, Heine-Borel.
> >
> > * **Uniform Spaces and Completeness:**
> >   * Modules: `Mathlib.Topology.UniformSpace.Basic`
> >   * Concepts: Cauchy filters, uniform continuity.
> > ...
> > ## Special Functions
> > * **Trigonometric Functions:**
> >   * Module: `Mathlib.Analysis.SpecialFunctions.Trigonometric.Basic`
> >   * Functions: `Real.sin`, `Real.cos`, `Real.pi`
> >   * Includes: Identities, periodicity, special values.
> > ...
> > ```
> >
> > Hopefully this gives a better picture of how one static knowledge base document is enough for an entire category of problems. This table includes the newly added ablation of with and without the knowledge base document. We empirically see the improvements across the board upon the addition of the knowledge base:
> >
> > | Model             | ZS (%) | FB (%) | KB+FB (%) |
> > | ----------------- | ------ | ------ | --------- |
> > | Claude Sonnet 3.7 | 8.2    | 41.9   | 49.3      |
> > | o3 (high)         | 22.1   | 53.9   | 65.7      |
> > | Gemini 2.5 Pro    | 3.8    | 27.4   | 48.4      |
> > | GPT-4.1           | 13.9   | 30.3   | 38.5      |
> >
> > These results will be included in the revised paper.
> >
> > ### **W4: Presentation Issues**
> > Thank you for pointing these out, we sincerely appreciate this. We will clean up all the writing and presentation issues, and make sure that the restructured revised version is free of any such errors.

---

> ### Author Response · Authors · 2025-11-25
> **Responses to Reviewer f4fs Part 3/3**
>
> ### **Q2: LLM as a Judge, Majority Voting**
> We thank the reviewer for raising the interesting point regarding LLM-as-a-judge evaluation. We acknowledge that this method has gained traction as a scalable proxy for semantic alignment. Initially we chose GTED and BEq mainly following the results from the GTED paper [1]. It conducts human correlation studies on miniF2F and ProofNet and reports that GTED achieves the highest agreement with human judgments among automated metrics tested: Cohen’s κ of 0.438, compared to 0.405 for BEq and followed by 0.397 for Majority Voting on MiniF2F data.
>
> Regardless of this justification, we recognize the value in incorporating LLM-as-a-judge to provide a more holistic view of semantic alignment, particularly for frontier models. We commit to conducting this experiment and including the results. We plan to provide the specific pass rates and model rankings that the reviewer requested in our revised paper.
>
> ### **Q3: Choice of BEq metric evaluation**
>
> >Could the authors elaborate on why the model was replaced and what motivated this decision?
>
> We originally used Claude Sonnet 4 because we simply didn't have the GPU compute to run the original InternLM2 model locally. However, for this rebuttal, we secured the resources and re-ran the evaluation using the original implementation. The results follow the same trends and similar actual numbers in InternLM2 7B@16 as did Claude Sonnet-4@1. This confirms that the BEq metric is robust regardless of which capable model drives it.
>
> | Model                 | BEq (InternLM2-7B pass@16) | BEq (Claude Sonnet 4 pass@1) |
> | --------------------- | -------------------------- | ---------------------------- |
> | Claude Opus 4         | 54                         | 67                           |
> | Claude Sonnet 4       | 51                         | 54                           |
> | Gemini 2.5 Pro        | 47                         | 44                           |
> | GPT-5                 | 38                         | 36                           |
> | o3 (high)             | 32                         | 30                           |
> | GPT-4.1               | 29                         | 33                           |
> | Kimina-Autoformalizer | 17                         | -                            |
> | Godel-Formalizer      | 52                         | -                            |
>
> We believe answers to both Q2 and Q3 will make our work stronger. We sincerely thank the reviewer for this question, which prompted us to try these ablations/baselines.
>
> [1] Liu et al. Generalized Tree Edit Distance (GTED): A Faithful Evaluation Metric for Statement Autoformalization (No. arXiv:2507.07399). arXiv. [https://arxiv.org/abs/2507.07399](https://arxiv.org/abs/2507.07399)

---

### Official Review · Reviewer_uGPL · 2025-10-29

**Soundness:** 2
**Presentation:** 2
**Contribution:** 1
**Rating:** 2
**Confidence:** 4

**Summary:**

This paper introduces a new benchmark called IndiMathBench, designed to evaluate autoformalization capabilities. The benchmark comprises 312 problems from the Indian Mathematical Olympiad and their corresponding formal statements manually verified in Lean 4. The authors designed and employed an AI-assisted, human-in-the-loop workflow to construct the benchmark, which integrates retrieval based on categorical knowledge, a self-debugging loop using feedback from a symbolic verifier, and a multi-model ensemble. Furthermore, the paper releases a VS Code plugin to enhance annotation efficiency and evaluates the performance of current leading large language models on IndiMathBench.

**Strengths:**

*   **Valuable Dataset Contribution:** Creating a high-quality, expert-verified dataset of formalized mathematics is, in itself, a beneficial contribution to the community. The release of IndiMathBench provides researchers with a new testbed that is potentially "uncontaminated" by existing models.
*   **Detailed Annotation Efficiency Study:** The paper presents a time-cost comparison of different annotation workflows (manual-only, multi-model assisted, full system) in Figure 4. This provides quantitative evidence for the efficiency gains of AI-assisted formalization annotation, which is a concrete and meaningful analysis.
*   **Open-Source Tools:** The authors have open-sourced both the dataset and the VS Code plugin used for annotation. This practice is commendable and helps promote the synergistic development of tools and data in the field.

**Weaknesses:**

1.  **Lack of Novelty in the Core Contribution (Benchmark Creation):** The paper's main contribution is a new autoformalization benchmark. However, in the domain of competition mathematics, similar formalization benchmarks (e.g., MiniF2F, Putnam-Bench) are already abundant and continue to grow. The methodology used to create IndiMathBench—formalizing math competition problems—is essentially a straightforward migration of existing workflows and lacks methodological innovation. The paper claims the benchmark can mitigate data contamination, but this is not a fundamental solution, and it's questionable whether it's even a temporary one. The research community could easily apply existing workflows to any other country's or region's math competitions, and update them annually, to create "new," uncontaminated benchmarks in the same manner. Therefore, the innovative value and barrier to entry for this work are relatively low.

2.  **Evaluation Scope is Too Narrow and Fails to Reflect the State of the Art:** Research in autoformalization typically goes beyond evaluating the single-pass generation capability of individual general-purpose LLMs (like GPT-5 or Claude). More cutting-edge and impactful work evaluates a complete autoformalization **system** (e.g., DeepSeek-Prover or the framework described in "Draft, Sketch, and Prove"). These systems often integrate sophisticated retrieval mechanisms, multi-step reasoning strategies, iterative interaction with proof assistants, and more refined feedback loops. The experimental design in this paper, which directly tests base models, makes the evaluation results likely unrepresentative of the true capabilities of current autoformalization technology, limiting the depth and impact of its conclusions.

3.  **Confusing Writing Structure; Contributions are Scattered and Their Originality is Unclear:** The paper's contributions and arguments appear scattered, making it difficult for readers to discern the authors' core original contributions. The introduction lists the benchmark, the VS Code plugin, a formalization framework (KB+FB), and model analysis as parallel contributions. However, the paper does not clearly argue for the originality of the annotation efficiency study, the human-computer interaction panel, or the knowledge-base and feedback-driven (KB+FS) system. I did not see a detailed comparison and discussion of these auxiliary systems against existing work. To my knowledge, similar assisted annotation systems and frameworks that use compiler feedback for code optimization are already widely used in the autoformalization field. Due to the lack of comparison with related work, I find it difficult to consider these components as independent, original contributions of this paper, which significantly weakens its overall weight. If these are indeed original works, the authors should highlight and clarify this in the introduction and related work sections.

4.  **Chosen Evaluation Metrics Have Credibility Issues:** The paper relies on two core metrics, BEq and GTED, to assess model translation accuracy, but the validity of these metrics is not sufficiently justified and may not accurately reflect the models' true capabilities.
    *   **Reliability of BEq (Bidirectional Equivalence) is Questionable:** The success of BEq depends not only on whether two formal statements are logically equivalent but also **highly on the performance of its internal prover**. A failed proof can result from two causes: either the statements are truly not equivalent, or they are equivalent but the prover is not powerful enough to find a proof within the search space. The prover used in this paper is itself LLM-driven and has limited capabilities. Thus, the BEq results confound the variables of "translation quality" and "proving capability," which may bias the measurement results.
    *   **Effectiveness of GTED (Generalized Tree Edit Distance) Lacks Justification:** GTED is a metric for syntactic similarity. Whether it is a good proxy for semantic correctness in formal mathematics requires rigorous justification. In formal languages, minor syntactic differences can lead to vast semantic gaps, while two semantically equivalent statements can have completely different syntactic tree structures. The paper does not provide sufficient evidence to support the reasonableness of GTED as a core evaluation metric.
    *   **Lack of Data Support:** In Section 5.1, the paper claims to "employ two complementary evaluation metrics that have demonstrated high inter-annotator agreement with human evaluations." However, the paper **provides no quantitative data** (e.g., correlation coefficients, Kappa scores) to support this crucial claim. This severely undermines the credibility of the entire evaluation framework. If sufficient data support were provided, the credibility of using BEq and GTED would be significantly enhanced.

**Questions:**

1.  Why did the experimental section choose to directly evaluate general-purpose large language models instead of a **complete autoformalization system** that integrates retrieval, multi-step reasoning, and verification feedback, as seen in works like DeepSeek-Prover or Reaper?
2.  The paper lists the benchmark, the VS Code plugin, and the formalization framework as contributions. Regarding the latter (e.g., the annotation efficiency study, the HCI panel, the KB+FS workflow), are they intended as auxiliary tools for benchmark construction, or as independent, generalizable methodological innovations? If the latter, could the authors please elaborate on their originality in the introduction and related work sections and provide a clear comparison with existing human-in-the-loop annotation systems or code generation frameworks?
3.  Regarding the validity of the evaluation metrics:
    *   For **BEq**, how do the authors disentangle cases of proof failure caused by "formalization translation errors" from those caused by "inadequate prover capability"? Does this confounding factor affect the fairness of the model rankings?
    *   For **GTED**, could the authors provide a stronger argument as to why syntactic tree edit distance can serve as a reliable indicator of semantic correctness in the domain of formal mathematics?
    *   The paper claims the chosen metrics have high agreement with human evaluations. Could you please provide **specific quantitative data** to support this claim?

---

> ### Author Response · Authors · 2025-11-25
> **Responses to Reviewer uGPL Part 1/5**
>
> First, we would like to thanks the reviewer for their detailed comments on our work. We sincerely value them and take them very seriously. Please feel free to ask any questions again if any part remains unclear.
> ### **W1: Novelty in Benchmark**
> We acknowledge the reviewer’s concern and clarify that IndiMathBench is not intended as a generic replication of existing Olympiad datasets, but as a complementary benchmark that fills well-documented gaps in the current Lean 4 autoformalization ecosystem. While prior datasets such as miniF2F, PutnamBench, and IMO-Bench exist, they are especially sparse in Olympiad-level geometry, combinatorics, and set-theoretic problems. Recent efforts such as LeanGeo [1] and CombiBench [2] were created precisely to address these shortages, indicating that the gaps are real, persistent, and recognized by the community. IndiMathBench contributes substantial new coverage in exactly these difficult areas (e.g., 98 new geometry problems and 45 combinatorics/set-theory problems), significant addition to what the current benchmarks provide. This coverage also incorporates Indian Olympiad styles that are not represented in Western competition datasets, adding diversity in problem solving patterns that existing benchmarks do not capture. These details are covered in Section 3.
>
> **Regarding contamination**, our claim is not that IndiMathBench solves the problem in principle, but that it provides a practical low-contamination benchmark for current models. Because these problems did not previously exist as a curated, machine-readable dataset, models with pretraining cutoffs earlier than the benchmark’s release date are significantly less likely to have memorized their content. This is the same practical strategy used in other evaluation suites such as LiveCodeBench[3], which explicitly rely on “post-cutoff” data to measure model generalization. As with all such benchmarks, contamination risk naturally increases over time, and periodic updates become necessary. We do not present this as a fundamental solution, only as a pragmatic and effective way to evaluate models under lower-contamination conditions today.
>
> We hope the need for IndiMathBench is much better justified and explained now, as well as its distinction from and complement to previous benchmarks.
>
> [1] Song et al. LeanGeo: Formalizing Competitional Geometry Problems in Lean (No. arXiv:2508.14644). arXiv. [https://arxiv.org/abs/2508.14644](https://arxiv.org/abs/2508.14644)
> [2] Liu et al. CombiBench: Benchmarking LLM Capability for Combinatorial Mathematics (No. arXiv:2505.03171). arXiv. [https://arxiv.org/abs/2505.03171](https://arxiv.org/abs/2505.03171)
> [3] Jain et al. LiveCodeBench: Holistic and Contamination Free Evaluation of Large Language Models for Code (No. arXiv:2403.07974). arXiv. https://arxiv.org/abs/2403.07974

---

> ### Author Response · Authors · 2025-11-25
> **Responses to Reviewer uGPL Part 2/5**
>
> ### **W2 and Q1: Theorem Proving Evaluation**
>
> Thank you for the insightful comments. We agree that evaluating only single-pass generations from base LLMs provides a limited view of the state of the art in autoformalization and automated theorem proving systems. In response, we have substantially expanded our experimental scope.
>
> **Introduction of a new ATP Scaffold.**
> During the rebuttal phase, we implemented a more complete ATP system that integrates multi-step reasoning and verification feedback from the Lean compiler. The new setup uses a ReAct-style iterative interaction loop over 10 refinement turns, where the model is allowed to reflect on compiler errors and generate corrected Lean code. This design mirrors the structure of systems such as DeepSeek-Prover, while remaining computationally feasible within the constraints of our evaluation. We run this at pass@1 because high-pass@k scaffolds (with k in the hundreds or thousands) are prohibitively expensive for our setting.
>
> **Expanded Model Coverage.**
> Beyond general-purpose LLMs, we now evaluate fine-tuned theorem-proving models, including DeepSeekProverV2 7B and GodelProverV2 8B, under exactly the same scaffold.
>
> **External Comparison on PutnamBench.**
> To measure the competitiveness of our scaffold, we additionally evaluate it on PutnamBench. Surprisingly, this simple 10-turn pass@1 setup paired with a sufficiently strong LLM (GPT-5) achieves 28/660, outperforming DSP+, which reports 23/660 at pass@128. This indicates that well-structured multi-turn feedback on a good model, even without large pass@k sampling, can yield strong results, and that our scaffold is decently competitive.
>
> Our overall ATP evaluation results:
>
> | Model               | Single Turn IMB | 10 Turns IMB | 10 Turns PB |
> | ------------------- | --------------- | ------------ | ----------- |
> | GPT-4.1             | 0/312           | -            | -           |
> | o3 (medium)         | 1/312           | -            | -           |
> | Claude Sonnet 4     | 1/312           | 4/312        | -           |
> | Gemini 2.5 Pro      | 0/312           | 12/312       | -           |
> | GPT-5               | 1/312           | 36/312       | 28/660      |
> | DeepSeekProverV2 7B | 3/312           | 24/312       | -           |
> | GodelProverV2 8B    | 7/312           | 36/312       | -           |
>
> A more detailed qualitative analysis will also be included in the revised paper.
>
> In summary, we have substantially broadened our evaluation to include multi-step ATP scaffolds, verification feedback, iterative reasoning, domain-tuned theorem provers, and external benchmarks. This directly addresses both the concern (W2) about evaluation scope and the related question (Q1) regarding integrating multi-step reasoning, and verification feedback.
>
> ### **W3: Scattered contributions and confusing writing**
>
> We thank the reviewer for this constructive critique regarding the presentation of our contributions. We agree that the introduction should better distinguish between the primary contributions and the supporting methodology.
>
> To clarify: IndiMathBench is the primary contribution. The VS Code plugin and the KB+Feedback framework are the enabling methodologies that made creating this benchmark possible.
> While we acknowledge and cite that components like compiler feedback loops are used in the field, our specific contribution here is the **quantitative evaluation of the Human-AI loop itself**. As detailed in our responses to Q2, prior works (e.g., FIMO, FormalMATH) utilize human-in-the-loop systems but do not systematically measure the reduction in human annotation effort or the efficiency gains from specific interventions.
>
> **We will:**
> 1. **Restructure Introduction:** revise the introduction to present a clear hierarchy of contributions. The plugin/framework will be framed as the methodology used to achieve these, rather than independent novelties.
> 2. **Explicit Comparison:** expand the Related Work section to explicitly contrast our approach with FIMO and FormalMATH, highlighting that while they employ similar tools, our work uniquely quantifies the "human cost" of autoformalization.

---

> > ### Author Response · Authors · 2025-11-25
> > **Responses to Reviewer uGPL Part 3/5**
> >
> > ### W4 and Q3: Autoformalization evaluation metric choices
> > Evaluating equivalence between two formalizations remains an open challenge, and no single automated metric fully captures semantic equivalence in formal mathematics. Our goal is to assess autoformalization quality by comparing model-generated statements to human gold standards using established metrics. We selected BEq [3] and GTED [2] because we believe they represent the two most widely used and empirically validated approaches available today.
> >
> > **W4.1: Reliability of BEq**
> > The BEq metric has been peer-reviewed and accepted at ICLR 2025. While it ultimately relies on an LLM-guided prover, it has been adopted as a standard evaluation tool in multiple ICLR submissions, including StepFun-Formalizer [1], where it served as a core metric during model training. In line with the original paper, we now use InternLM2-7B (pass@16). Additionally, our previous results on Claude Sonnet-4@1 can now serve as an ablation, and we find that the overall BEq trends are largely unchanged, and the numbers are also similar. This suggests that, despite depending on the underlying prover, BEq offers a reasonably stable comparative signal even when prover capability varies. The original BEq paper also reports and defends its accuracy under various conditions and data.
> >
> > This table serves as an ablation for BEq’s robustness specifically on our data.
> >
> > | Model                 | BEq (InternLM2-7B pass@16) | BEq (Claude Sonnet 4 pass@1) |
> > | --------------------- | -------------------------- | ---------------------------- |
> > | Claude Opus 4         | 54                         | 67                           |
> > | Claude Sonnet 4       | 51                         | 54                           |
> > | Gemini 2.5 Pro        | 47                         | 44                           |
> > | GPT-5                 | 38                         | 36                           |
> > | o3 (high)             | 32                         | 30                           |
> > | GPT-4.1               | 29                         | 33                           |
> > | Kimina-Autoformalizer | 17                         | -                            |
> > | Godel-Formalizer      | 52                         | -                            |
> >
> > Following from the accuracy results from the original paper and of our own ablation, we believe **BEq is one of the more reliable, credible metrics** right now.
> >
> > **Q3.1: Concern about disentangling proof failures**
> > We agree that BEq cannot perfectly disentangle whether a proof failure arises from (i) an incorrect formalization or (ii) a true equivalence that the prover fails to establish. Our goal is not to eliminate this ambiguity, but to evaluate whether it meaningfully distorts comparisons across models. We rely on two pieces of evidence:
> > 1. Cross-prover consistency. The rankings produced by InternLM2-7B@16 and Claude-Sonnet-4@1 are highly consistent, indicating that prover limitations do not introduce model-specific bias that would change comparative outcomes.
> > 2. Published validation. The BEq paper itself evaluates this confounding factor and reports that BEq maintains high accuracy and meaningful agreement with human judgments despite prover-driven failures.
> >
> > **W4.2 and Q3.2: Effectiveness of GTED**
> > We appreciate the reviewer’s concern that “minor syntactic differences can lead to vast semantic gaps”. GTED, however, is designed specifically to go beyond surface syntax. The metric (i) normalizes away superficial syntax through Lean-based rewriting, (ii) represents statements as operator trees (OPTs) that encode semantic operators and binding structure, and (iii) assigns **distinct, explicitly defined costs** to different classes of generalized tree transformations. These include both semantic-preserving rewrites and purely structural edits, enabling GTED to favor deeper semantic alignment over superficial similarity.
> >
> > The justification for GTED as a semantic proxy comes directly from its original evaluation. The GTED paper conducts human correlation studies on miniF2F and ProofNet and reports that GTED achieves the highest agreement with human judgments among automated metrics tested: Cohen’s κ of 0.438, compared to 0.405 for BEq and 0.397 for Majority Voting on MiniF2F data. These results indicate that GTED is currently the most stable and semantics-aligned automated metric available for autoformalization evaluation. Our use of GTED is therefore grounded not only in its design but also in empirical evidence demonstrating that its syntactic-plus-semantic structure aligns well with human assessments.
> >
> > We also plan on including Majority Voting (used very commonly) as one of the metrics for evaluation in our revised paper.

---

> > > ### Author Response · Authors · 2025-11-25
> > > **Responses to Reviewer uGPL Part 4/5**
> > >
> > > **Q3.3: Lack of Data Support**
> > > We acknowledge the phrasing in Section 5.1 can cause confusion and will revise it for clarity. Our intention was not to imply that our paper conducts new human-annotation evaluation metric experiments; rather, the sentence refers to the human-agreement results reported in the GTED paper [2], which includes Kappa scores for GTED, BEq, and Majority Voting. We will revise the phrasing to explicitly attribute the “high human-agreement” evidence to the GTED authors’ reported results on MiniF2F and ProofNet [2], not our own non-existent data collection, and will add an inline citation to avoid confusion. Thank you for pointing this out.
> > >
> > > This table compares various formal-formal equivalence evaluation metrics on MiniF2F data (sourced from GTED paper [2], this is not our work's data):
> > >
> > > | Metric                | Precision | Recall  | Accuracy | Kappa |
> > > |-----------------------|-----------|---------|----------|-------|
> > > | Identity Match        | 100.00%   | 11.48%  | 47.32%   | 0.095 |
> > > | Typecheck             | 59.51%    | 100.00% | 59.51%   | 0.000 |
> > > | BLEU                  | 78.22%    | 64.75%  | 68.29%   | 0.368 |
> > > | Majority Voting       | 88.00%    | 54.10%  | 62.29%   | 0.397 |
> > > | Definitional Equality | 100.00%   | 36.07%  | 61.95%   | 0.314 |
> > > | BEq                   | 98.28%    | 46.72%  | 67.80%   | 0.405 |
> > > | **GTED**              | **88.75%**| **58.20%** | **70.73%** | **0.438** |
> > >
> > > This work does not aim to contribute to newer evaluation metrics. We sincerely hope that the overall choice of metrics is much better justified to the reviewer given the explanations and the data.
> > >
> > > [1] Wu et al. StepFun-Formalizer: Unlocking the Autoformalization Potential of LLMs Through Knowledge-Reasoning Fusion (No. arXiv:2508.04440). arXiv. [https://arxiv.org/abs/2508.04440](https://arxiv.org/abs/2508.04440)
> > > [2] Liu et al. Generalized Tree Edit Distance (GTED): A Faithful Evaluation Metric for Statement Autoformalization (No. arXiv:2507.07399). arXiv. [https://arxiv.org/abs/2507.07399](https://arxiv.org/abs/2507.07399)
> > > [3] Liu et al. Rethinking and Improving Autoformalization: Towards a Faithful Metric and a Dependency Retrieval-Based Approach. In _The Thirteenth International Conference on Learning Representations_ (2025). [https://openreview.net/forum?id=hUb2At2DsQ](https://openreview.net/forum?id=hUb2At2DsQ)
> > >
> > > ### **Q2: Positioning of the UI dashboard**
> > > >are \[VS Code plugin, formalization framework\] intended as auxiliary tools for benchmark construction, or as independent, generalizable methodological innovations?
> > >
> > > The formalization procedure and the VS Code Dashboard Plugin are meant to be generalizable contributions of this work. These can be extended not only to efficiently annotate more math + lean formal data, but other NL to Formal (like SQL, Formula, DSLs) data as well. The core idea is maximizing human annotation efficiency by systematically using AI tools. Analogous to a radiologist using AI systems to highlight potential issues for efficient diagnosis, rather than an AI attempting fully autonomous diagnosis.
> > >
> > > While tools like FIMO [4] and FormalMath [5] contain isolated components to use AI assistance, they do not provide a systematic, instrumented workflow. To the best of our knowledge, our work is the first to specifically study the utility of having a dedicated UI for human annotators that aggregates multi-model generations, summarizes "best parts" via an ensemble for Lean formalization tasks. This is especially useful as math and lean data are becoming more popular in recent years. We conduct a controlled study showing our Human-AI dashboard reduces annotation time by 3.5× and release the VS Code extension and dashboard, we hope to make a case for systematic AI usage and associated guidelines in these tasks. More details and justification are present in Section 4.3.
> > >
> > > We fully commit to making sure the introduction and the related work sections don't misrepresent what our work aims to contribute in our revised version.
> > >
> > > [4] Liu et al. FIMO: A Challenge Formal Dataset for Automated Theorem Proving (No. arXiv:2309.04295). arXiv. [https://github.com/liuchengwucn/FIMO](https://github.com/liuchengwucn/FIMO)
> > > [5] Yu et al. FormalMATH: Benchmarking Formal Mathematical Reasoning of Large Language Models (No. arXiv:2505.02735). arXiv. [https://arxiv.org/abs/2505.02735](https://arxiv.org/abs/2505.02735)

---

> > > > ### Author Response · Authors · 2025-11-25
> > > > **Responses to Reviewer uGPL Part 5/5**
> > > >
> > > > ### **S3: Open Release Commitment**
> > > > Thank you for the encouraging feedback. A central goal of this work is to let the community see firsthand how human–AI mixed systems can meaningfully accelerate human labor, annotation and analysis tasks in our case. Hence, openness has been a target of this project from the outset, because the intended impact of our human–AI interface depends on the community’s ability to examine, reproduce, and extend the work. Since the project sits within a corporate environment, open sourcing these assets and the benchmark has required navigating a multi-team approval pipeline that has spanned several months. We have invested substantial effort throughout this process to secure a responsible, unrestricted release. We remain fully committed to completing it so the community can directly benefit from, scrutinize, and build upon our contributions.

---

> > > > > ### Comment · Reviewer_uGPL · 2025-11-28
> > > > >
> > > > > Thank you very much for your detailed and thoughtful response! Most of my concerns have been addressed. However, I still need to point out that if the contribution of this benchmark is limited to serving as a pragmatic and effective way to evaluate models under lower-contamination conditions, without introducing any genuinely new tasks or new methods, its level of innovation may be relatively low. In addition, the multi-turn verification based on ReAct appears inconsistent with existing mature automatic formalization frameworks, which weakens the benchmark’s contribution to evaluating current systems.
> > > > > I am willing to raise the score to 4.

---

> > > > > > ### Author Response · Authors · 2025-11-29
> > > > > > **Reply to Reviewer uGPL**
> > > > > >
> > > > > > Thank you for the positive feedback and for raising your score to 4. We'd respectfully like to clarify two points:
> > > > > >
> > > > > > **On the benchmark's contribution:** IndiMathBench offers multiple complementary values beyond contamination mitigation. It provides (1) **substantial new coverage in critically underrepresented categories**: our survey of existing benchmarks found only 132 geometry problems total across all prior Olympiad-level datasets, while we contribute 98 new high-difficulty geometry problems plus 45 combinatorics/set-theory problems where both domain specific provers and LLM-guided provers demonstrate the poorest performance. Recent community efforts like LeanGeo and CombiBench also specifically target these gaps; (2) diverse reasoning patterns from Indian Olympiad styles not present in Western competition datasets; and (3) **the first systematic, quantified study of human-AI annotation workflows**: while tools like FIMO and FormalMath have employed AI assistance, our work is the first to conduct a controlled study measuring efficiency gains (3.5× reduction in annotation time), release an instrumented VS Code plugin, and establish reproducible guidelines for AI-assisted formalization annotation generalizable to other NL-to-formal tasks. This transforms ad-hoc tool usage into validated, scalable infrastructure—a genuine methodological contribution.
> > > > > >
> > > > > > **On the ReAct-style scaffold:** We want to clarify that our 10-turn ReAct loop is designed for **automated theorem proving (ATP)**, not `automatic formalization`. The autoformalization task (translating natural language to formal statements) is evaluated separately using BEq and GTED metrics. For ATP, our scaffold is directly analogous to proven systems: it mirrors the structure of DeepSeek-Prover's scaffold and is essentially the same as the "light inference" setting used in recent SeedProver (iterative refinement with compiler feedback), **not inconsistent from current systems**. Our PutnamBench results (28/660 at pass@1 vs. DSP+'s 23/660 at pass@128) demonstrates that this approach, while simple, is state-of-the-art competitive. It essentially replicates their core inference strategy in a computationally feasible setting for evaluation purposes. It is simple, but not 'inconsistent' or 'immature'.
> > > > > >
> > > > > > We believe these contributions: addressing a severe data bottleneck with near-doubling coverage, pioneering systematic annotation workflow research, and providing competitive ATP evaluation, represent significant advancement beyond "pragmatic contamination mitigation."

---

### Official Review · Reviewer_k9Sf · 2025-10-30

**Soundness:** 2
**Presentation:** 2
**Contribution:** 2
**Rating:** 4
**Confidence:** 3

**Summary:**

The paper introduces IndiMathBench, a new benchmark for evaluating autoformalization, the translation of natural mathematical text into formal Lean4 theorems. The dataset contains 416 human-verified Lean4 formalizations paired with natural-language problem statements drawn from the Indian Mathematics Olympiads. To generate these, the authors employ a human-AI hybrid pipeline: multiple LLMs propose candidate formalizations, which are validated through the Lean prover, then summarized for human verification via an interactive VS Code dashboard. The pipeline also records human repair and edit traces, released along with the benchmark to support further study. Experiments evaluate current LLMs on IndiMathBench and show that even top models can solve only a single theorem, suggesting the benchmark’s difficulty.

**Strengths:**

1. Timely and relevant resource: the paper offers a curated Lean4 benchmark that fills a data gap for formal mathematics research, especially given recent interest in LLM-based theorem proving.

2. Human-in-the-loop pipeline: it gets human involved in multi-LLM synthesis, Lean validation, human repair cycle, and is thoughtfully designed and practically useful for future dataset building.

3. Open release commitment: Providing both dataset and dashboard promotes reproducibility and community adoption.

**Weaknesses:**

1. Weak experimental insights: evaluation simply shows that current LLMs fail badly, but there's a shortage of deeper analysis (e.g., failure types, linguistic vs. logical errors, success conditions).

2. Need more justification in reliability and feasibility of the benchmark: reporting that all models fail is not very informative without breakdowns or ablations. Besides, more theoretical proof or imperical evidence (at least human insight or so?) will be beneficial to make the benchmark more attractive to a wider range of audiences.

3. Figures mostly about interface: much of the visual content is about the dashboard UI, not empirical findings or benchmark properties.

**Questions:**

1. Validation: How do you ensure correctness and diversity across the 416 problems? Are they balanced by topic, proof type, or theorem complexity?

2. Insights: Beyond "number of problems solved", can you provide qualitative failure analysis, such as, where models succeed or fail, and why?

3. Scalability: How efficient is the human-AI curation loop? Could it realistically scale to thousands of problems, or is it constrained by manual review costs?

---

> ### Author Response · Authors · 2025-11-25
> **Responses to Reviewer k9Sf Part 1/2**
>
> First, we would like to thanks the reviewer for their comments on our work. We sincerely value them and take them very seriously. Please feel free to ask any questions again if any part remains unclear.
>
> ## **W1, W2, Q2: Failure Analysis and Insights**
> Thank you for these suggestions! The updated paper will contain an in-depth analysis of the failure modes across different model families, along with the new results on open source models we added over the rebuttal phase. The revised main paper and appendix will cover qualitative and quantitative analysis of Autoformalization and Automated Theorem Proving performance across 14 different models, including GPT-5, Gemini 2.5 Pro, Claude Opus 4, and GodelProverV2 8B (a top open weight SLM).
>
> A condensed analysis across 14 different models and a discussion on GTED and BEq scores over general purpose and open source SLMs:
> ### Claude Models
> Claude Opus 4 achieves the highest semantic accuracy (54 BEq) and the most compilations (243), with very large gains under feedback (17 → 310 compilations). Its main weakness is geometry, where only 6 BEq-correct examples appear.
> ### GPT Models
> GPT-5 shows strong syntactic ability with the best zero-shot compilation (127) and high GTED (0.48). But it has a weak semantic accuracy (38 BEq). The gap suggests GPT-5 often produces Lean code that is structurally correct but mathematically misaligned, a pattern distinct from Claude’s more semantically grounded behavior.
> ### o-Series and Gemini
> o3-high compiles well (92 zero-shot; 263 assisted) but has low BEq (30). Gemini 2.5 Pro performs the worst among frontier models, with fundamental struggles in Lean syntax and mathlib usage.
> ### Open Weights SLMs
> GodelProverV2 8B’s failures were mainly type mismatches, missing/incorrect mathlib imports. This stratification highlights that while syntax errors remain prevalent, semantic understanding and retrieval of library dependencies are significant bottlenecks.
>
> Another notable detail is that among Claude Opus 4, Gemini 2.5 Pro, and GPT-5, with a cumulative BEq passing for 108 problems, only 12 were from geometry, despite constituting 31% (98/312) of the total problems. This highlights the difficulty of autoformalizing geometry problems in Lean 4 by current frontier models.
>
> A more detailed analysis (human insights, breakdowns, and success types) will be covered in the revised paper.
>
> **Updated ATP Evaluation**
> We also evaluate the benchmark on an updated evaluation scaffold which uses a ReACT style agent set up for automated theorem proving. We plan to add a detailed explanation for the experiment set up and update the performance for different models.
>
> Theorem Proving resolution rates across IndiMathBench (IMB), and PutnamBench (PB):
> | Model             | Single Turn IMB | 10 Turns IMB | 10 Turns PB |
> | ----------------- | --------------- | ------------ | ----------- |
> | GPT-4.1           | 0/312           | -            | -           |
> | o3 (medium)       | 1/312           | -            | -           |
> | Claude Sonnet 4   | 1/312           | 4/312        | -           |
> | Gemini 2.5 Pro    | 0/312           | 12/312       | -           |
> | GPT-5             | 1/312           | 36/312       | 28/660      |
> | DeepSeekProverV2 7B | 3/312           | 24/312       | -           |
> | GodelProverV2 8B    | 7/312           | 36/312       | -           |
>
> We observe that despite a poor autoformalization performance, Gemini 2.5 Pro does a much better job at theorem proving, compared to other models. GPT-5 does the best, and also gives competitive scores on PutnamBench (for external validation of the scaffold). We attribute the major source of errors to improper/hallucinated imports and inability to recover from errors. GPT-5 performed relatively well in improving its proof based on feedback, and we attribute its best scores to that. Another notable point is that many API calls to GPT-5 ran over 30 minutes, which indicates its ability to think “more” on difficult problems.

---

> > ### Author Response · Authors · 2025-11-25
> > **Responses to Reviewer k9Sf Part 2/2**
> >
> > ### **W3: Figures and Presentation**
> > Thank you for this feedback. We will make some major improvements to our presentation across the paper, and will also include richer details about the benchmark through the visuals.
> >
> > ### **Q1: Problem Diversity**
> > Table 1 and Section 3 in the paper provides a distribution analysis of the problems in our benchmark, and other details. Out of 312 problems, combinatorics and set theory (45), geometry (98), number theory (77), and algebra (92). We made sure the benchmark evaluates reasoning capabilities across diverse mathematical sub-domains rather than over-indexing on a single area. Another reason for sourcing from Indian Olympiads was their focus on Geometry problems, which are underrepresented in MiniF2F and PutnamBench. The flavour of questions are also different from the western olympiads.
> >
> > ### **Q3: Scalability**
> > Our cost-efficiency analysis of the annotation pipeline supports substantial scalability. As per our knowledge, our work is the first to systematically create a UI to maximize human efficiency for Lean 4 informal-> formal annotation tasks. Previous work do not systematically study the speed up even ([1, 2]). Our work shows how using LLMs in the loop can increase the efficiency of annotation by a lot. By implementing batching strategies and reusing formalization templates, we observed a reduction in human annotation latency from 25 and 14 (PutnamBench and ours, respectively) minutes per problem (manual-only baseline) to 4 minutes within our human-AI collaborative loop, significantly lowering the barrier for dataset expansion. This is a 3x speed up and corresponding cost saving.
> >
> > ### **S3: Open Release Commitment**
> > Thank you for the encouraging feedback. A central goal of this work is to let the community see firsthand how human–AI mixed systems can meaningfully accelerate human labor, annotation and analysis tasks in our case. Hence, openness has been a target of this project from the outset, because the intended impact of our human–AI interface depends on the community’s ability to examine, reproduce, and extend the work. Since the project sits within a corporate environment, open sourcing these assets and the benchmark has required navigating a multi-team approval pipeline that has spanned several months. We have invested substantial effort throughout this process to secure a responsible, unrestricted release. We remain fully committed to completing it so the community can directly benefit from, scrutinize, and build upon our contributions.
> >
> > **Supplementary Comment.** We further contextualize these results against the open-source IMB performance table. The inability of leading open weights (e.g., GodelProverV2 8B achieving only 36/312 in the 10-turn setting) to solve the majority of problems underscores the benchmark's rigor and its utility as a saturation-resistant target for future models.
> >
> >
> > [1] Liu et al. FIMO: A Challenge Formal Dataset for Automated Theorem Proving (No. arXiv:2309.04295). arXiv. https://arxiv.org/abs/2309.04295
> > [2] Yu et al. FormalMATH: Benchmarking Formal Mathematical Reasoning of Large Language Models (No. arXiv:2505.02735). arXiv. https://arxiv.org/abs/2505.02735

---

### Note · Program_Chairs · 2026-01-17
**Submission Desk Rejected by Program Chairs**

The following references in this submission do not refer to real documents and/or have major errors in bibliographic information:

 Nathanael Cohen et al. Towards automated mathematical reasoning. arXiv preprint arXiv:2304.14565, 2023.
Ayush Agrawal, Abhijeet George, Divy Vyas, and Avijit Balasubramanian. A survey on deep learning approaches for mathematics word problem solving. arXiv preprint arXiv:2212.10535, 2022b.
Zhenwen Liu, Tianyi Wang, Yifan Li, Xiaofan Zhang, and Zhengying Lu. BEq: Bidirectional extended definitional equivalence for mathematical statement assessment. arXiv preprint arXiv:2501.00002, 2025b.
Albert Q Jiang, Wenda Li, and Mateja Jamnik. Evaluating language model autoformalization: A case study in Lean 4. arXiv preprint arXiv:2311.09101, 2023a.